# Increased CO$_2$ fixation enables high carbon-yield production of 3-hydroxypropionic acid in yeast

Ning Qin [1,9], Lingyun Li[1,2,9], Xiaozhen Wan[1], Xu Ji[1], Yu Chen [3], Chaokun Li [4], Ping Liu[5], Yijie Zhang[1], Weijie Yang[1], Junfeng Jiang[6], Jianye Xia [6], Shuobo Shi [1], Tianwei Tan [1], Jens Nielsen [1,2,7] ✉, Yun Chen [2,8] ✉ & Zihe Liu[1] ✉

CO$_2$ fixation plays a key role to make biobased production cost competitive. Here, we use 3-hydroxypropionic acid (3-HP) to showcase how CO$_2$ fixation enables approaching theoretical-yield production. Using genome-scale metabolic models to calculate the production envelope, we demonstrate that the provision of bicarbonate, formed from CO$_2$, restricts previous attempts for high yield production of 3-HP. We thus develop multiple strategies for bicarbonate uptake, including the identification of Sul1 as a potential bicarbonate transporter, domain swapping of malonyl-CoA reductase, identification of Esbp6 as a potential 3-HP exporter, and deletion of Uga1 to prevent 3-HP degradation. The combined rational engineering increases 3-HP production from 0.14 g/L to 11.25 g/L in shake flask using 20 g/L glucose, approaching the maximum theoretical yield with concurrent biomass formation. The engineered yeast forms the basis for commercialization of bio-acrylic acid, while our CO$_2$ fixation strategies pave the way for CO$_2$ being used as the sole carbon source.

Biobased chemical[1] and biofuel[2] production as replacement of petroleum-based production could assist in mitigating green-house gas emissions while establishing sustainable economic activities[3]. However, these biobased productions also result in emissions of the greenhouse gas CO$_2$, and fixing emitted CO$_2$ into products is therefore of great interest[4], in particular as this may also result in higher carbon-yields and therefore improves process economics.

Intracellular inorganic carbon exists in four forms: dissolved CO$_2$, carbonic acid, bicarbonate, and carbonate[5], that could be interconverted based on the pH and salinity. It has been reported that compared with 5% CO$_2$ supply, 10% CO$_2$ could increase itaconic acid production by two-fold in yeast[6]. At pH 7.4 and 20 °C, the concentration of CO$_2$ in equilibrium with the gaseous state is only 0.012 mM, whereas the concentration of bicarbonate in equilibrium with air is 0.26 mM[7]. Since the pH in the cytoplasm is approximately neutral[8], inorganic carbon mainly exists in the form of bicarbonate[9], which means increasing intracellular bicarbonate concentrations could be more appealing than enhancing the higher CO$_2$ partial pressure. Bicarbonate can be captured by native carboxylation reactions, of which the acetyl-CoA carboxylation catalysed by acetyl-

[1]College of Life Science and Technology, Beijing Advanced Innovation Center for Soft Matter Science and Engineering, Beijing University of Chemical Technology, Beijing 100029, China. [2]Department of Life Sciences, Chalmers University of Technology, SE412 96 Gothenburg, Sweden. [3]Key Laboratory of Quantitative Synthetic Biology, Shenzhen Institute of Synthetic Biology, Shenzhen Institute of Advanced Technology, Chinese Academy of Sciences, Shenzhen 518055, China. [4]Stem Cells and Metabolism Research Program, Faculty of Medicine, University of Helsinki, 00014 Helsinki, Finland. [5]The State Key Laboratory of Chemical Resource Engineering, College of Chemical Engineering, Beijing University of Chemical Technology, Beijing 100029, China. [6]Tianjin Institute of Industrial Biotechnology, Chinese Academy of Sciences, Tianjin 300308, China. [7]BioInnovation Institute, Ole Maaløes Vej 3, DK2200 Copenhagen, Denmark. [8]Novo Nordisk Foundation Center for Biosustainability, Technical University of Denmark, 2800 Kongens, Lyngby, Denmark. [9]These authors contributed equally: Ning Qin, Lingyun Li. ✉e-mail: nielsenj@chalmers.se; yunc@chalmers.se; zihe@mail.buct.edu.cn

CoA carboxylase Acc1 is known to carry a high flux in yeast. The generated malonyl-CoA can be transformed into 3-hydroxypropionic acid (3-HP), which represents a valuable chemical intermediate. Efficient capture of $CO_2$ into 3-HP can improve the carbon yield in its production, and this step is also crucial in the native $CO_2$ fixation pathway 3-hydroxypropionate/4-hydroxybutyrate (HP/HB) cycle[10] and 3-hydroxypropionate bi-cycle[11].

3-HP holds a significant position as the potentially third-largest biobased platform compound, with an estimated potential annual market size of USD 10 billion[12,13]. One of the key reasons behind its prominence is its ease of dehydration into acrylate, a crucial component used in the production of synthetic polymers like super-absorbents applied in disposable diapers[14]. The production of 3-HP has thus garnered considerable attention, but the major hurdle remains the low yield of 3-HP on the substrate, posing a significant challenge to establish an industrial scale process[15]. A comprehensive analysis of 3-HP production in yeast reveals that the best yield achieved thus far in terms of the 3-HP to substrate (glucose) ratio is approximately 0.31 g/g[16], which corresponds to only 39.7% of the maximum theoretical yield[15]. This falls short of the commercial requirement for 3-HP to acrylate production, which highlights the current limitations in 3-HP production and underscores the need for significant improvements in yield to meet the demands of the industry.

In this study, we conduct production envelope analysis by Yeast8 model and find that previous attempts to produce 3-HP is restricted by the availability of bicarbonate. We therefore engineer yeast to break this restriction and hereby achieve high carbon yield production of 3-HP. This is achieved through employing multiple strategies to increase the accessible bicarbonate, minimize native $CO_2$ release, as well as by avoiding carbon waste. Hereby we achieve a 0.5625 g 3-HP/g glucose yield (89.3% of the theoretical yield of 3-HP with biomass generation). Furthermore, the high titre of 3-HP obtained through a microbial fermentation can easily be transformed to acrylate with a yield of 100%.

## Results

### Limited provision of bicarbonate restricts 3-HP production

The production envelope analysis of 3-HP via the malonyl-CoA reductase pathway was calculated using the Yeast8 model[17], and this was used to analyse the carbon flux and identify limiting factors for 3-HP production. Contrary to the general belief that the precursors of acetyl-CoA and malonyl-CoA, as well as the redox cofactors and ATP are the key requirements for 3-HP production, our analysis suggested that a high flux of bicarbonate (Fig. 1a) could be crucial for efficient 3-HP production, i.e., the flux of bicarbonate should be 1.562 times higher than that of the substrate (glucose) to achieve the maximum 3-HP production (Fig. 1a). Unfortunately, such a flux of bicarbonate is unattainable within the native metabolism, even if we assume that bicarbonate can be freely converted from $CO_2$. The cellular metabolism is simply incapable of providing enough $CO_2$ to generate the required bicarbonate for 3-HP production[18], regardless of the metabolic state being respiration (efficient $CO_2$ generation state) or fermentation (non-efficient $CO_2$ generation state) (Fig. 1a). Moreover, if we consider that the $CO_2$ generated is in a gaseous state intracellularly, it would promptly leave the liquid phase, and the carboxylation process by Acc1 necessitates the presence of bicarbonate dissolved in water (Fig. 1b). These factors further exacerbate the deficiency of bicarbonate. Specifically, in the native metabolic context, the limited flux of bicarbonate significantly reduced the theoretical maximum yield of 3-HP from 0.781 g/g glucose to approximately 0.440 g/g (fermentation state)−0.490 g/g (respiration state) (Fig. 1a). This result indicated that the availability of bicarbonate is the primary bottleneck for 3-HP production, effectively restricted all previous attempts at metabolic flux rewiring in this production envelope boundary. To address this issue, it is imperative to ensure an ample supply of

bicarbonate for Acc1 carboxylation while minimizing $CO_2$ loss during acetyl-CoA and energy generation.

### The carbon conserved loop from glucose to 3-HP

We started by establishing a carbon conserved loop that effectively recycled the lost carbon to increase the theoretical carbon-yield for 3-HP production (Fig. 1b). The split mutant of the bi-functional enzyme MCR from *Chloroflexus aurantiacus*[19] was used for 3-HP production, as it exhibited efficient conversion of malonyl-CoA to malonic semi-aldehyde (MSA) via the mutated MCR-C$^{N941V\,K1107W\,S1115R}$ domain and then to 3-HP via the MCR-N domain[20] (Fig. 1b). Integration of the split MCR into the chromosome together with *ACC1* overexpression resulted in a 3-HP titre of 0.136 g/L in the QLW1 strain (Fig. 1c). Then, to test our hypothesis we added 10 mM sodium bicarbonate to the medium, and indeed increased the 3-HP titre to 0.15 g/L (*P* value = 0.012813). Thus, for the following fermentation processes, 10 mM sodium bicarbonate was consistently added unless described otherwise.

Next, to reduce the $CO_2$ release while producing acetyl-CoA, we integrated the phosphoketolase (PK) pathway reported in our previous study[21], including a xylulose-5-phosphate-prefered phosphoketolase (xPK) from *Leuconostoc mesenteroides* and a phosphotransacetylase (PTA), from *Clostridium kluyveri*, and enhanced 3-HP production to 0.39 g/L (Fig. 1c, QLW2). Further, to release Acc1 from post-translational regulation, we introduced two site mutations, S659A and S1157A, which prevent phosphorylation of Acc1 by Snf1, thereby activating Acc1's activity[22]. It then resulted in an increased 3-HP titre to 0.65 g/L in the QLW3 strain (Fig. 1c).

To achieve a higher flux through the PK pathway, extensive genotype modifications were reported to redirect carbon flux from glycolysis to the oxidative pentose phosphate (oxPP) pathway[21]. We instead turned to transcription factors (TFs) to investigate whether a single gene modification could achieve the overall carbon flux rewiring. Stb5 is a TF and an NADPH regulator[23] that represses the expression of *PGI1*, coding the rate-limiting reaction of glycolysis at glucose-6-phosphate to rewire the flux from glycolysis to the oxPP pathway[24]. Stb5 has been used to redirect the carbon flux for the production of protopanaxadiol in yeast[25]. We thus overexpressed *STB5* with the *TEF1* promoter. However, the growth of the strain was significantly inhibited (Supplementary Fig. 1). qPCR results indicated that the strength of the *TEF1* promoter was around 12-fold higher than that of the native *STB5* promoter (Fig. 1d, QLW4), that may have been too strong for regulating the TF expression in the cell. Thus, we placed the TATA box adjacent to the core promoter of *TEF1*[26] (Fig. 1d) and constructed an artificial promoter to replace the native promoter of *STB5*, and qPCR results showed that its strength was around 2.5-fold greater than that of the native *STB5* promoter (Fig. 1d, QLW5). Indeed, the strain with this artificial promoter-controlled *STB5* enhanced 3-HP production to 0.74 g/L (Fig. 1c, QLW5).

Meanwhile, we tried to employ carbonic anhydrase and a bicarbonate transporter to enhance the intracellular bicarbonate content for the production of 3-HP (Fig. 1e). This $CO_2$ concentrating system has successfully increased the production of succinate in *E. coli*[27,28]. Similarly, overexpression of *NCE103*, the native carbonic anhydrase in yeast[29], could also improve 3-HP production to 0.7 g/L in QLW6 (Fig. 1f). Furthermore, we tried to find a native bicarbonate transporter in yeast. Bicarbonate diffuses poorly into the cell[27], and therefore requires SLC4 and SLC26 as two kinds of anion transporters to assist the active transport process[30]. Bor1, a boron efflux transporter of the plasma membrane belonging to the SLC4 family, can also bind bicarbonate (Fig. 1g)[31], whereas no bicarbonate transporter from the SLC26 family has been reported in yeast[32]. To identify potential bicarbonate transporters, we performed a phylogenetic analysis of key species of the SLC4 and SLC26 families (Fig. 1g). We identified Sul1/2, which were previously regarded as sulphate permeases[33]. Bor1 is an efflux transporter, while Sul1/2 are influx transporters. Yeast Sul1 and Sul2 share a

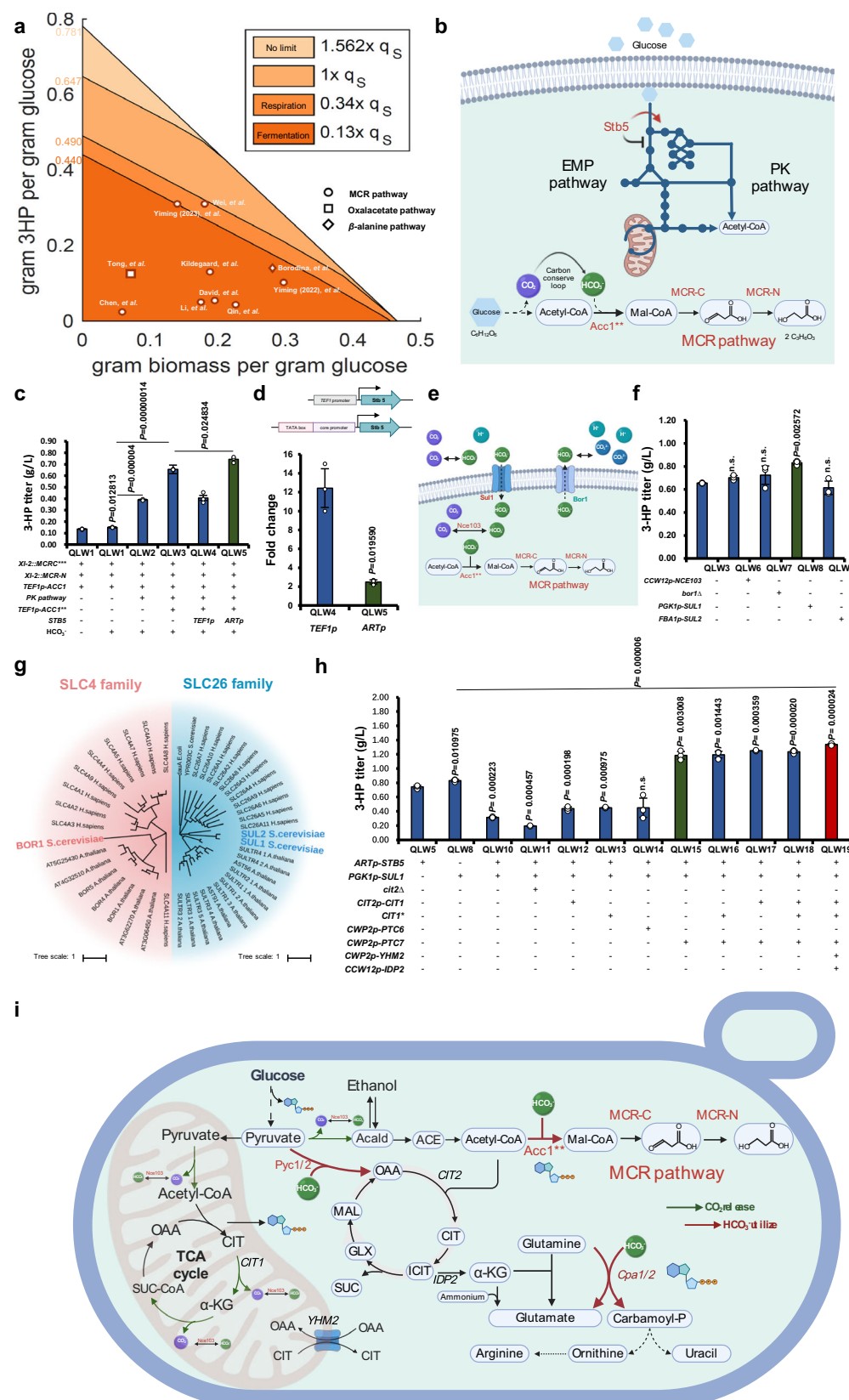

close phylogenetic relationship with human SLC26A11, a $Cl^-/HCO_3^-$ exchanger in the kidney[34]. While both Sul1 and Sul2 structures exhibit significant similarity to SLC26A11, Sul1 displays a higher number of equivalent positions (545 equivalent positions), suggesting a higher possibility for the potential bicarbonate import function of yeast Sul1 (Supplementary Fig. 2). To verify this, we overexpressed *SUL1*, *SUL2*,

and deleted *BOR1*, respectively. Hereby the 3-HP titre increased to 0.83 g/L in *SUL1* overexpressing yeast, which was a 27.7% improvement over the control strain (Fig. 1f, QLW8). Meanwhile, the final $OD_{600}$ of this strain also increased from 6.97 to 7.65, which was 9.8% higher than that of the control strain (Supplementary Fig. 3). These results demonstrated the potential of Sul1 as the bicarbonate importer and

**Fig. 1 | Production envelope analysis and bicarbonate metabolism rewiring for CO$_2$ fixation. a** The production envelope analysis for 3-HP production. Different colour represented the range of 3-HP production with its concurrent biomass under different bicarbonate flux. Reports used the malonyl-CoA pathway[16,21,22,102–106], the oxaloacetate pathway[44], and the β-alanine pathway[82] were labelled with dot, square, and diamond. The q$_S$ indicated the consumed rate of glucose. The DCW was either measured directly or estimated based on OD$_{600}$ (DCW equals to 70% of OD$_{600}$). **b** The malonyl-CoA reductase pathway was used to produce 3-HP. Transcription factor Stb5 could realize the carbon flux rewiring from glycolysis to the oxPP pathway. **c** Adjusting *STB5* expression by *ARTp* could together with the expression optimization of MCR domains, Acc1 and the PK pathway enhance 3-HP production to 0.74 g/L. **d** Substitution of the native promoter of *STB5* with the *TEF1*p and an artificial promoter *ARTp*, respectively, and quantification of the *STB5* with *TEF1*p and *ARTp* strength by qPCR. **e** Cellular bicarbonate derived from the potential bicarbonate transporter Sul1 (into the cell), Bor1 (outside the cell), and the endogenous synthesis from CO$_2$ by native carbonic anhydrase Nce103. **f** Providing more bicarbonate by overexpressing *SUL1* enhanced 3-HP production to 0.83 g/L. **g** Phylogenetic analysis of SLC4 and SLC26 families identified Bor1, Sul1, and Sul2 as potential bicarbonate transporters in *S. cerevisiae*. (**h**) Rewiring the carbon flux from OAA to the high production of 3-HP. Overexpressing the phosphatase Ptc7 could significantly increase the 3-HP production based on introducing Cit1$^{S462A}$ mutation, overexpressing *CIT1, YHM2*, and *IDP2*. (**i**) Distribution of CO$_2$ releasing reactions and bicarbonate utilizing reactions in *S. cerevisiae*. Abbreviations were defined in Supplementary Data 5 and 6. 10 mM sodium bicarbonate was added to the defined minimal medium was used for the 3-HP production. All data were presented as mean ± SD of biological triplicates. Significant comparisons of two groups were indicated in the graphs statistical analysis performed using a two-tailed unpaired Student's *t*-test (*$p < 0.05$, **$p < 0.01$, ***$p < 0.001$). Source data are provided as a Source Data file.

the importance of intracellular bicarbonate availability for 3-HP production.

Next, we intended to combine the above strategies of *STB5* and *SUL1* overexpression to close this carbon conservation loop, yet the 3-HP production in the resulted QLW10 strain dropped to 0.31 g/L, which was even lower than that of the reference strains QLW5 and QLW8 (Fig. 1h). Analysis of extracellular metabolites revealed the presence of 0.37 g/L of oxaloacetate (OAA) in the QLW10 strain but not the QLW8 strain (Supplementary Fig. 4). We speculated that the increased intracellular bicarbonate content also improved other bicarbonate utilizing reactions, such as the carboxylation of pyruvate to OAA catalysed by Pyc1 and Pyc2, and therefore tried various strategies to decrease the accumulation of OAA (Fig. 1i). Yet, the deletion of *CIT2* that encoded the key gene of the glyoxylate cycle didn't recover 3-HP titre (Fig. 1h, QLW11), with acetate accumulated instead (Supplementary Fig. 5). We next focused on enhancing the utilization of OAA. Firstly, we overexpressed *CIT1* encoding the mitochondrial citrate synthase that converts OAA and acetyl-CoA to citrate, and improved 3-HP production to 0.44 g/L (Fig. 1h, QLW12). Similarly, we tested Cit1$^{S462A}$ enabling the enzyme to defer phosphorylation regulation[35], and increased 3-HP production to 0.45 g/L (Fig. 1h, QLW13). We also overexpressed Ptc6 and Ptc7, which could selectively dephosphorylate mitochondrial proteins, especially the Ptc7 that could dephosphorylate Cit1 and enhance yeast respiration[36]. Indeed, overexpression of *PTC7* in the QLW15 strain could not only recover but further increase the production of 3-HP to 1.18 g/L (Fig. 1h). Next, we overexpressed Yhm2, an OAA transporter in the mitochondrial membrane[37], and the cytosolic NADP-specific isocitrate dehydrogenase Idp2 that could provide more cytosolic NADPH[37], further increased 3-HP production to 1.34 g/L (Fig. 1h, QLW19), that were 81.1% and 60.9% improvement over the control strain QLW5 and QLW8, respectively.

## Improved conversion of malonyl-CoA to 3-HP

To provide enough malonyl-CoA for 3-HP production, another copy of *ACC1*$^{S659A\ S1157A}$ with *TEF1* promoter was integrated into the chromosome, yet the growth of the strain was strongly disturbed. It has been reported that the intracellular malonyl-CoA level is strictly regulated by the AMP-activated kinase (AMPK)[38] signalling pathway and a high concentration of malonyl-CoA could be toxic to the cell[39]. This result suggested that the elevated malonyl-CoA concentration being beyond the range the cell could handle[40]. Therefore, we focused on more efficient conversion of malonyl-CoA into 3-HP.

We started by expressing MCR-N and MCR-C in different combinations, and expressing the MCR-N domain and MCR-C$^{N941W\ K1107W\ S1115R}$ domain[16,20] using a higher copy number plasmid increased 3-HP production to 4.12 g/L in QLW23 (Fig. 2a, b). Currently, there is no crystal structure of MCR from *C. aurantiacus*, and only crystal structures of the MCR-N domain and the MCR-C domain from *Porphyrobacter dokdonensis* have been published[41]. Using SWISS-MODEL, we generated a homologous model of MCR from *C. aurantiacus* using known structures of the MCR-N domain (PDB: 6k8v) and MCR-C domain (PDB: 6k8s) from *P. dokdonensis* (Fig. 2c) as the template. We noticed that the MCR-N domain could be separated into two short-chain dehydrogenase/reductase[41] domains: SDR domain 1 (MCR-N domain/SDR1, M1–L284) and SDR domain 2 (MCR-N domain/SDR2, P285–I550). The MCR structure revealed that the cofactor and substrate only bind to the SDR1 domain, indicating that enzyme catalysis may not occur at the SDR2 of the MCR-N domain[41]. Yet, when we deleted the MCR-N-SDR2 domain, the production of 3-HP had a strong decrease (Fig. 2d, QLW24). This result suggested that even though the MCR-N-SDR2 domain has no catalytic function, it may help the MCR-N-SDR1 domain to maintain its proper structure. To find a more efficient domain converting MSA to 3-HP, we performed domain swapping using SDRs from different species and obtained the improved malonyl-CoA reductase. Specially, we identified potential MCR-N-SDR1 domains through the substrate structural similarity, such as Ora1 (reported as the serine dehydrogenase)[42] from *S. cerevisiae*, and YdfG (reported as the NADP$^+$-dependent dehydrogenase with broad substrate specificity on 3-hydroxy acids)[42] from *E. coli* (Fig. 2c). Serine can be seen as 2-amino-3-hydroxypropionic acid, which could be regarded as the amino group substituted at the second carbon atom site of 3-HP. The substrate structural similarity together with the broad substrate specificity to 3-hydroxy acids of Ora1 and YdfG indicated the potential of these enzymes for 3-HP production. Indeed, the use of Ora1 and YdfG as MCR-N-SDR1 exhibited 26.1% and 28.5% improvements, producing 5.2 g/L and 5.3 g/L 3-HP, respectively (Fig. 2d, QLW25, QLW26). Neither codon optimization nor the application of a protein linker[43] could further improve the production of 3-HP (Fig. 2d). The finding that common cell factories such as *S. cerevisiae* and *E. coli* possess native enzymes catalysing malonic semialdehyde to 3-HP could potentially explain why adjusting the expression the of MCR-N domain and MCR-C domain was always irregular and been impute into ambiguity explainations[20,44].

Meanwhile, since malonyl-CoA was also utilized for fatty acid synthesis in the cell[45], we tried to reduce the generation of fatty acids to increase the production of 3-HP (Fig. 2e). Specially, we used the *COX9* promoter, which is a weaker promoter than the native *FAS1* promoter, and *HXT1* promoter, which is repressed by glucose, to control the expression of *FAS1*, respectively[46] (Fig. 2f). Using this approach, we were able to increase the titre of 3-HP to 5.87 g/L (Fig. 2f, QLW30). Meanwhile, we also up-regulated the β-oxidation pathway to decompose fatty acids back to NADPH and acetyl-CoA for 3-HP production (Fig. 2e). The resulting the QLW33 strain overexpressing *POX1* that encodes fatty-acyl coenzyme A oxidase[47] improved 3-HP production to 5.8 g/L. Similarly, the QLW34 strain overexpressing *POX2* that encodes 3-hydroxyacyl-CoA dehydrogenase and enoyl-CoA hydratase[47], improved 3-HP production to 5.92 g/L (Fig. 2f). We thus combined these strategies, yet did not observe a superimposed effect, with the 3-HP production in QLW36 fluctuated around 6 g/L (Fig. 2f). To verify the repression of lipid synthesis, we used Nile red that

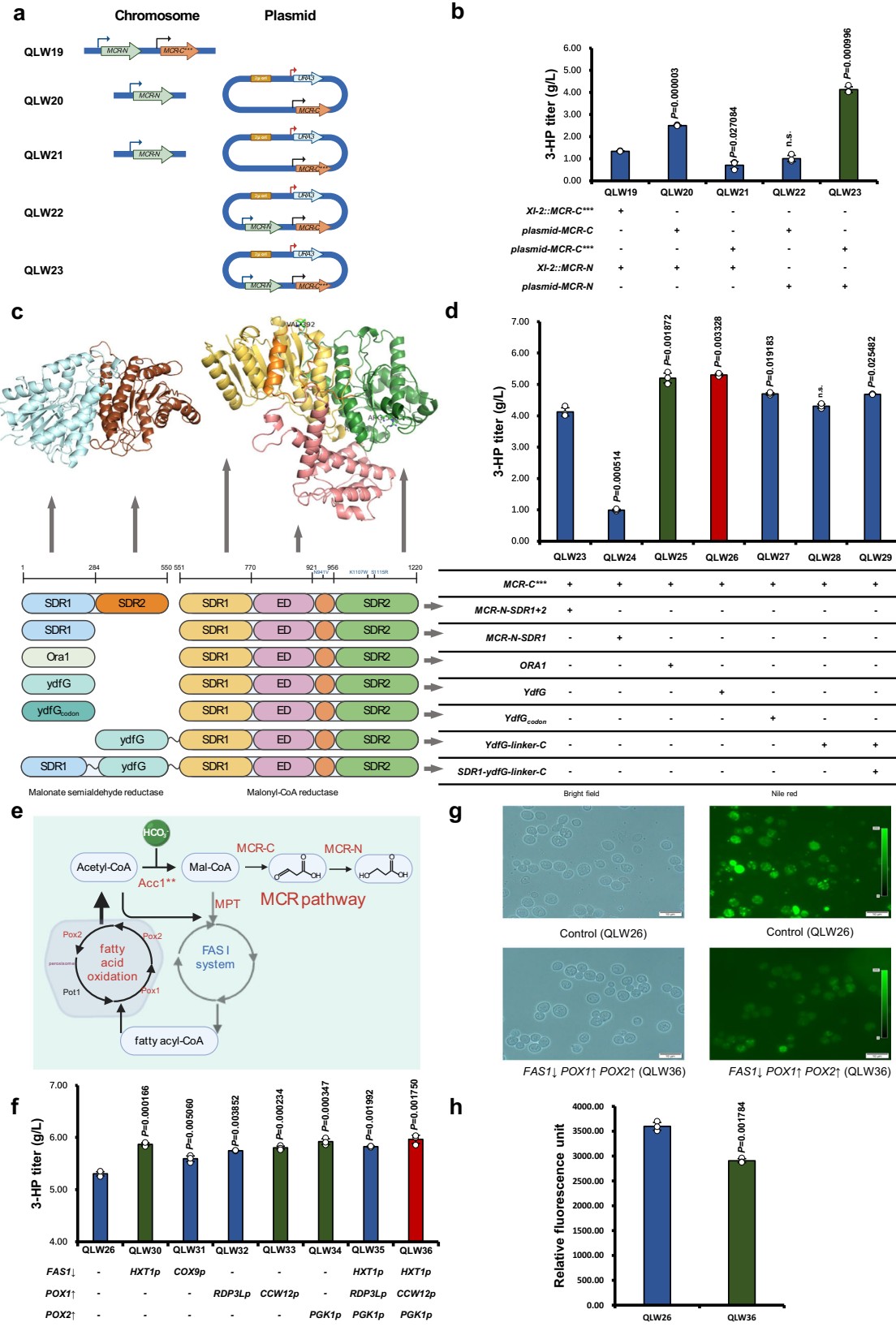

specifically binds to neutral lipids to characterize the yeast. The results showed a significant reduction in neutral lipids in the QLW36 strain compared with that of the control strain QLW26 (Fig. 2g, h), indicated a successful rewiring of the carbon flux away from fatty acids. That the titre of 3-HP could not be significantly increased may be because of other bottlenecks in the metabolism.

## Increasing 3-HP tolerance and the identification of a permease for transport of 3-HP

The 3-HP transport and toxicity represent a bottleneck for further improving 3-HP production in the QLW36 strain. Our experiments showed that after 72 h of fermentation with 5.5–6 g/L of 3-HP generated (strains QLW30 to QLW36), the pH of the medium dropped

**Fig. 2 | Enzyme and pathway engineering for efficient production of 3-HP.**
**a** Genes encoding of split MCR enzymes were either integrated in the chromosome or expressed using the plasmid system. **b** MCR-N domain and MCR-C*** domain co-expressed with the high copy plasmid could produce 4.12 g/L 3-HP. **c** Homologous modelling structure of the domains in malonate semialdehyde reductase and malonyl-CoA reductase from *C. aurantiacus*. **d** Swapping of native MCR domains with Ora1 and YdfG improved 3-HP production to 5.2 g/L and 5.3 g/L, respectively. **e** Carbon flux rewiring from fatty acids to 3-HP. **f** Downregulation of *FAS1* combined with upregulation of *POX2* and *POX1* improved the production of 3-HP. **g** Nile red staining demonstrated that the size and the number of lipid droplets decreased in the fatty acid oxidation strain QLW36 compared with that of QLW26. (**h**) The

fluorescence intensity of neutral lipids stained with Nile red in QLW26 and QLW36 was quantified in Relative Fluorescence Units (RFU). The RFU values were corrected by subtracting both the inherent autofluorescence of the samples and the fluorescence contributed by the solvent in the presence of Nile red (blank). Abbreviations were defined in Supplementary Data 5 and 6. The defined minimal medium with 10 mM NaHCO₃ was used for the 3-HP production. All data were presented as mean ± SD of biological triplicates. Significant comparisons of two groups were indicated in the graphs statistical analysis performed using a two-tailed unpaired Student's *t*-test (*$p < 0.05$, **$p < 0.01$, ***$p < 0.001$). Source data are provided as a Source Data file.

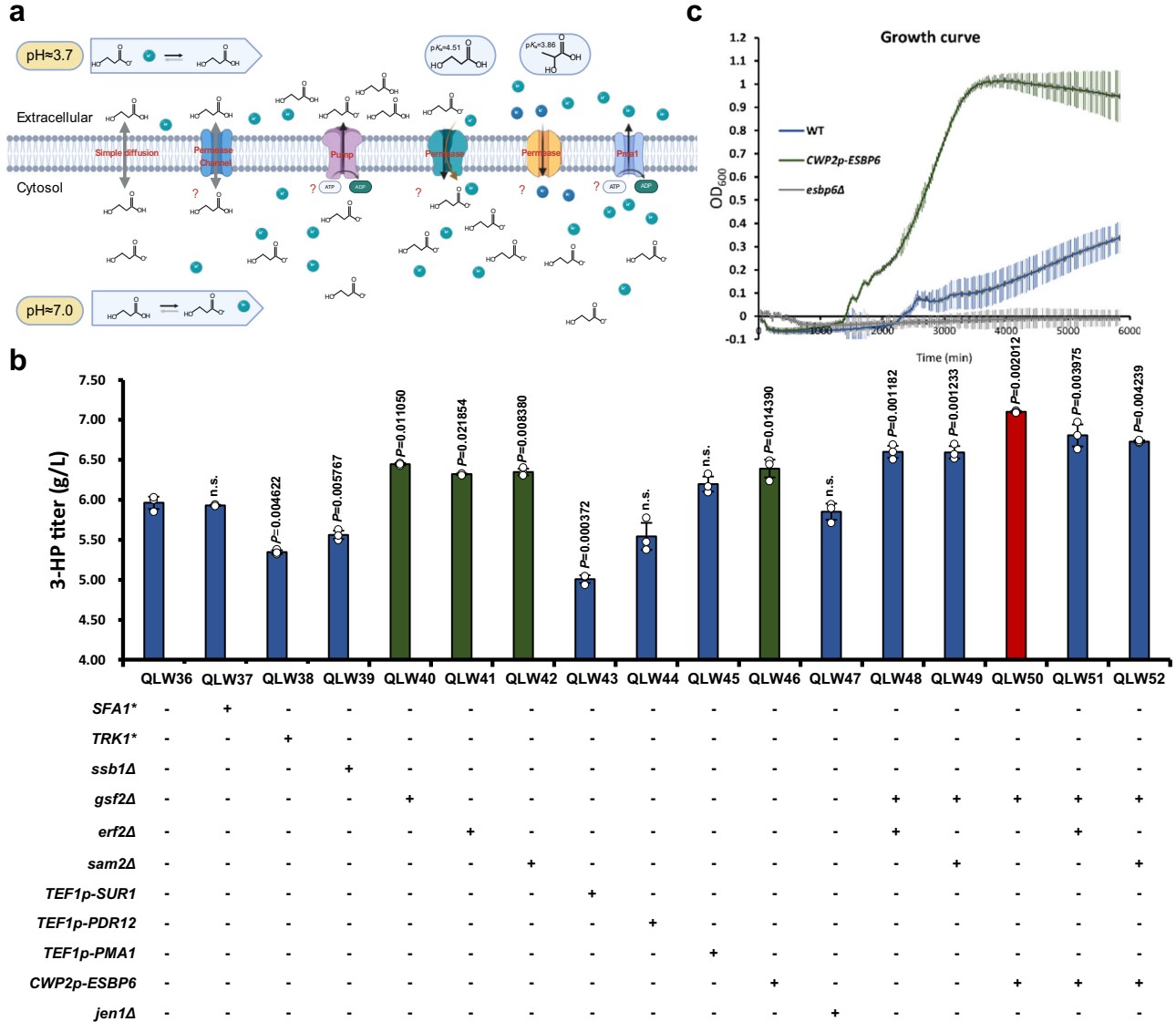

**Fig. 3 | Genotype identification of 3-HP tolerance and increase in the production of 3-HP. a** Schematic representation of transporters and permeases in the cell membrane. **b** Overexpression of *ESBP6*, deletion of *SAM2*, *GSF2*, or *ERF2* could significantly improve the production of 3-HP. **c** Growth profiling showed that overexpression of *ESBP6* could remarkably increase cell tolerance to high concentrations of 3-HP, compared with the WT strain. Abbreviations were defined in

Supplementary Data 5 and 6. The defined minimal medium with 10 mM NaHCO₃ was used for the 3-HP production. All data were presented as mean ± SD of biological triplicates. Significant comparisons of two groups were indicated in the graphs statistical analysis performed using a two-tailed unpaired Student's *t*-test (*$p < 0.05$, **$p < 0.01$, ***$p < 0.001$). Source data are provided as a Source Data file.

below 4. The pKa value of 3-HP is 4.51, and under this pH value, 3-HP remains predominantly undissociated in the medium (Fig. 3a). Free undissociated 3-HP can easily penetrate the cell membrane through diffusion and then release protons to become anions in the cytoplasm where the pH is around neutral[8]. The charged nature of the

anion cannot diffuse back out of the cell, resulting in its accumulation in the cytosol that could cause intracellular acidification, which is toxic for the cell[48]. The proton and anion must be pumped out of the cell, and this process is ATP-dependent through the H⁺-ATPase and ATP-binding cassette (ABC) pumps[48]. This process consumes a

large amount of ATP and can result in ATP depletion, leading to a decrease in the overall 3-HP yield.

We started by testing mutations to improve yeast tolerance of 3-HP (Fig. 3b). For example, the Sfa1 bifunctional alcohol dehydrogenase and the Sfa1[C276S] mutant have been reported to improve 3-HP tolerance in *S. cerevisiae*[49]. Trk1, a component of the Trk1-Trk2 potassium transport system, was reported to improve yeast tolerance to propionate by increasing potassium influx through the Trk1[A1169E] mutation[50]. Yet, these two mutations could not increase the production of 3-HP (Fig. 3b, QLW37 and QLW38). Moreover, lactate (pKa 3.86) and 3-HP (pKa 4.51) are structural isomers that have been found to exhibit the same degree of growth inhibition at the same undissociated free acid concentration[51,52], indicating that the mutation involved in the tolerance of lactate may have the potential to improve 3-HP tolerance. The deletion of the ribosome-associated chaperone Ssb1 was reported to improve yeast tolerance to lactate[53]. The deletion of *GSF2* could increase lactate production[54] by releasing the Crabtree effect of yeast[55], leading to upregulation of gene expression involved in respiration, which has a higher ATP yield than fermentation and is crucial to organic acid production[56]. Deletion of *ERF2*, a subunit of a palmitoyl transferase, could increase lactate production but not lactate tolerance in yeast[57]. Deletion of *SAM2*, which catalyses the synthesis of the cellular cofactor S-adenosylmethionine involved in phospholipid biosynthesis and membrane remodelling during acid stress, was also reported to increase lactate production in yeast[58]. Furthermore, the Sur1[I245S] mutation and the overexpression could increase lactate tolerance and production[57]. Indeed, we demonstrated that the deletion of either *GSF2* (QLW40), *ERF2* (QLW41), or *SAM2* (QLW42) significantly improved the production of 3-HP, to the levels of 6.45, 6.32, and 6.35 g/L, respectively (Fig. 3b).

So far, no 3-HP exporter has been reported. Based on the fact that lactate and 3-HP are structural isomers, we next turned to the identification of potential 3-HP exporters based on previous lactate reports. Plasma membrane ABC transporter Pdr12, which could transport carboxylic acids with an aliphatic chain ranging from one to seven[59] at the plasma membrane. P2-type H[+]-ATPase Pma1[60], which controls the intracellular pH, have been reported to improve lactate tolerance in yeast[61]. The overexpression of *ESBP6*, a protein with similarity to monocarboxylate permeases, was reported to improved lactate resistance and production in *S. cerevisiae*[62]. Meanwhile, the deletion of Jen1, the monocarboxylate/proton symporter in the plasma membrane[63], could increase the production of lactate in yeast[54,64,65]. Indeed, compared with that of the control strain QLW36, the overexpression of *ESBP6* improved the production of 3-HP to 6.39 g/L (Fig. 3b, QLW46). To verify the potential mechanism by which Esbp6 improved 3-HP production, *ESBP6* was overexpressed in the wild type (WT) strain, and could significantly increase cell tolerance compared with WT in a medium containing 50 g/L 3-HP at pH 3.5 (buffered with potassium citrate), while the deletion of *ESBP6* almost completely abolished growth in this medium (Fig. 3c). Moreover, the overexpression of *ESBP6* in the WT strain led to a notable reduction in intracellular 3-HP content compared with that of the WT strain (Supplementary Fig. 6). These results indicated that Esbp6 could help exporting 3-HP out of the cell, thereby increasing the tolerance of yeast and improving 3-HP production. To our knowledge, this could be the evidence for potential 3-HP exporters. Notably, combining the overexpression of *ESBP6* with the deletion of *GSF2* improved 3-HP production to 7.1 g/L in strain QLW50 (Fig. 3b).

### Conserved bicarbonate pool for 3-HP production

There are mainly three enzymes utilizing bicarbonate in yeast (Fig. 1i): Acc1 fixes bicarbonate with acetyl-CoA to form malonyl-CoA, Pyc1/2 fixes bicarbonate to transform pyruvate into OAA, as well as Cpa1/2 fixes bicarbonate to generate carbamoyl-phosphate[66], and then to the pyrimidine ring and arginine (Fig. 4a). It has been demonstrated that

bicarbonate limitation in the cytoplasm could restricted de novo nucleotide synthesis, which inhibits growth of cancer cells (the Weinberg effect)[67]. This report enlightens that the potential competition for bicarbonate between the de novo synthesis of the pyrimidine ring and malonyl-CoA may reduce 3-HP production (Fig. 4a). The plasmid used to express *MCR-C* and *YdfG* had a high copy number with a *URA3* marker. This means that the strain with this plasmid will have a stronger expression of *URA3*, a key step in pyrimidine ring synthesis, leading to the potential competition with Acc1 for bicarbonate. To conserve bicarbonate to 3-HP production, we replaced the *URA3* marker with the *HIS3* marker that is not involve in the bicarbonate metabolism[68] (Fig. 4a). Indeed, the *HIS3* marker plasmid significantly enhanced 3-HP production from 7.1 g/L to 8.22 g/L (Fig. 4b, QLW53), while the qPCR result even exhibited a moderate decrease in the plasmid copy number upon the change in the marker gene (Supplementary Fig. 7). To demonstrate the improvement of 3-HP production by auxotrophic marker swapping was caused by the decrease in the competition of bicarbonate for the de novo uracil synthesis, we added isotope labelled NaH[13]CO$_3$ to the medium and tested the M + 1 presence of uracil in strains with different auxotrophic-marker plasmids. Indeed, consistent with the significantly increased 3-HP production, the M + 1 presence of uracil was significantly decreased in the QLW53 strain compared with that of the QLW50 strain (Fig. 4b, c). These results confirmed that the auxotrophic marker swapping increased the 3-HP production by releasing the 3-HP generation from the bicarbonate competition with the de novo uracil synthesis (Fig. 4c). Different auxotrophs have been shown to impact yeast metabolism[69], which can affect the production of fatty acids[70]. Yet, mechanisms behind this impact have not been fully understood[70]. Here, we demonstrated that the distinct bicarbonate metabolic backgrounds of these auxotrophic marker turbulent metabolic capacities of yeast[71], thereby increasing the production of 3-HP.

The results of the enhanced 3-HP production with the *HIS3* marker plasmid suggested that cytoplasmic bicarbonate generated from dissolved CO$_2$ could be limited due to its low solubility. To provide enough bicarbonate for malonyl-CoA generation, we supplemented the medium with 75 mM (0.15 g/flask) calcium carbonate, which further enhanced 3-HP production from 8.22 to 9.12 g/L (Fig. 4d and Supplementary Fig. 8). To test whether the increase in 3-HP production was due to the addition of bicarbonate rather than pH buffering, we conducted a production of 3-HP experiment with 2-(*N*-morpholino) ethane sulfonic acid (MES) that could buffer the pH to the same level as calcium carbonate did. The results showed that the titre of the fermentation with MES buffering was only 8.16 g/L, which was not significantly different from the control (Fig. 4d). We have also tested different concentrations of calcium hydroxide, and only 40 mM (0.0592 g/flask) calcium hydroxide could slightly improve 3-HP production to 8.41 g/L (Supplementary Fig. 9), whereas when further increased calcium hydroxide concentrations the cells stopped growth. A similar strategy has been utilized to increase succinate production in industry, yet the mechanism underneath was not indicated[72,73]. Here by model calculation and experiment demonstration, we suggested that it was because the limited intracellular CO$_2$ concentration as well as the thermodynamic and kinetic favouring of the bicarbonate fixation reaction that increased the production.

### Inositol phosphate application provides more ATP for 3-HP production

The positive effect on 3-HP production by deleting *GSF2* indicated that the energy was crucial for the 3-HP yield. If we consider the energy demand for 3-HP transmembrane during production, this ATP deficiency would be more intense. We thus selected different strategies to provide more ATP for 3-HP production. *S. cerevisiae* has the Crabtree effect, which produces an excess amount of ethanol in high

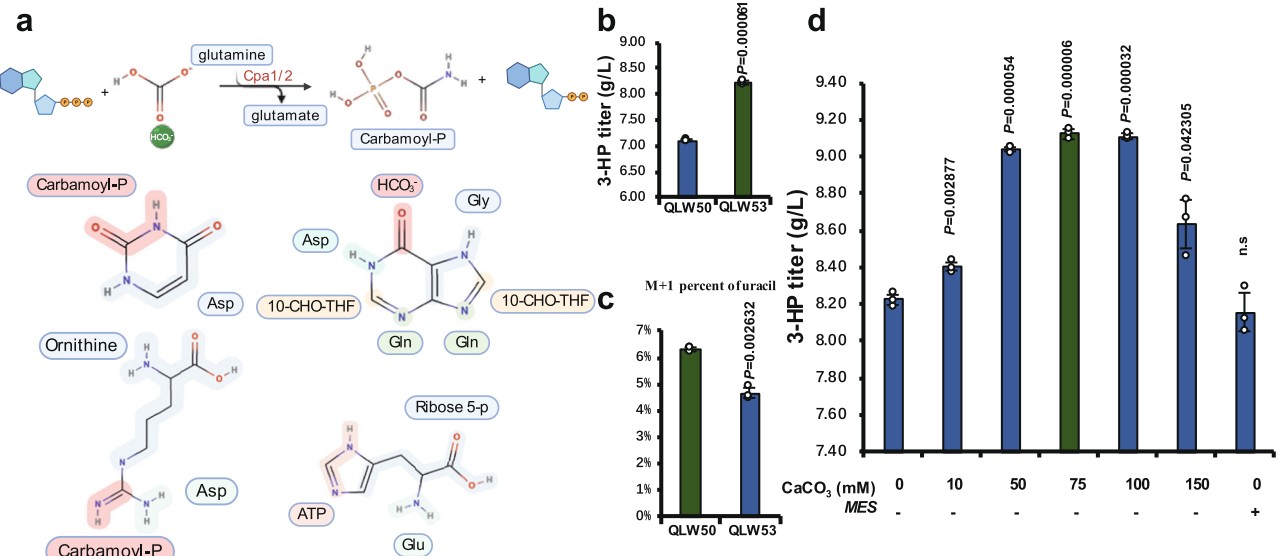

**Fig. 4 | Alleviating bicarbonate competitions from the de novo pyridine ring synthesis enhanced 3-HP production. a** Carbamoyl-P was synthesized by Cpa1/2 from ATP, bicarbonate, and glutamate. Atom sources for purine, pyrimidine, arginine, and histidine were marked. **b** Substitution of *URA3* marker by *HIS3* marker in the same plasmid background increased the 3-HP titre to 8.22 g/L in the QLW53 strain. **c** $^{13}C$ isotope labelling indicated that less M + 1 uracil was produced in QLW53 strain corresponding to the higher 3-HP titre. **d** CaCO$_3$ was used to provide more bicarbonate recycling CO$_2$, 0, 10, 50, 75, 100, and 150 mM represent 0 g/flask,

0.02 g/flask, 0.10 g/flask, 0.15 g/flask, 0.20 g/flask, and 0.30 g/flask. Abbreviations were defined in Supplementary Data 5 and 6. The defined minimal medium with 30 mM $^{13}C$ isotope labelling NaHCO$_3$ or the corresponding concentrations of CaCO$_3$ as shown in the figures was used for the 3-HP production. All data were presented as mean ± SD of biological triplicates. Significant comparisons of two groups were indicated in the graphs statistical analysis performed using a two-tailed unpaired Student's *t*-test (*$p < 0.05$, **$p < 0.01$, ***$p < 0.001$). Source data are provided as a Source Data file.

concentrations of glucose and reduces energy efficiency[74,75]. Moreover, the P/O ratio of *S. cerevisiae* is around 1, which indicates that one mole of glucose could only generate approximately 16 moles of ATP (oxidation phosphorylation plus substrate phosphorylation)[76]. Thus, rebalancing central metabolism and enhancing the ATP yield from glucose could be important.

After glucose is transported inside the cell, it would be phosphorylated by Hxk2 or Glk1 and Hxk1, which restricted its transportation out of the cell[77] (Fig. 5a). Deleting main functional carrier *HXK2* can reduce the glucose consumption rate, which could release the Crabtree effect to get a higher ATP yield[78]. Similarly, Mth1 is involved in the glucose signalling pathway, and its mutant Mth1$^{\Delta231}$ acts as a transcription factor to repress hexose transporters (Fig. 5a), that ultimately restricts the Crabtree effect and derepress respiration[78]. Meanwhile, the mutation of Med2*432Y, which is the tail module of RNA polymerase II mediator, can decrease the expression of genes involved in glycolysis and increase the expression of genes involved in protein synthesis. These mutations in principle may be able to release more ATP[79], yet did not increase the production of 3-HP in strain QLW53 any further (Fig. 5b). The glucose repression on the respiratory pathway may still be too strong that glycolysis and respiration may not be well balanced for high 3-HP production. We next turned to our recently identified Oca5 that degrade 5-diphosphoinositol 1,2,3,4,6-pentakisphosphate (5-InsP$_7$)[78], which serves as an energy sensor to sense ATP concentrations and balances gene expressions in glycolysis and respiration. Indeed, *OCA5* deletion significantly increased 3-HP production from 9.12 g/L in QLW53 to 10.15 g/L in QLW58 (Fig. 5b). To verify that the production enhancement was because of the rebalance of glycolysis and respiration rather than simply enhanced respiration, we tested the cultivation in baffled shake flask that could provide more oxygen for oxidative phosphorylation, and indeed the 3-HP production did not improve for QLW53 (Fig. 5b). We also tested the deletion of *OCA3*[80] and upregulation of *KCS1*[81], respectively, that could also increase 5-InsP$_7$ concentrations, yet they did not further improve 3-HP production in QLW58.

## The dual nature of Uga1 for 3-HP production

The structural similarity between 3-HP and serine allowed us to identify the serine dehydrogenase Ora1 for 3-HP production (Fig. 2d), while the isomerized relation of lactate and 3-HP allowed us to identify the potential 3-HP permease Esbp6 in yeast (Fig. 3b). Thus, we hypothesized that these similarities may also allow native enzymes to mediate the degradation of 3-HP or its intermediates (Fig. 6a). We started by deleting lactate dehydrogenases Dld1, Dld2, and Dld3, respectively, yet none of the deletions increased 3-HP production (Fig. 6b).

Moreover, it was demonstrated that the β-alanine-pyruvate aminotransferase (BAPAT) and gamma-aminobutyrate[82] transaminase Uga1 exhibited substrate promiscuity and could convert β-alanine to MSA for 3-HP production[82]. The transamination reaction is highly reversible[83], while studies have also used BAPAT to convert MSA to β-alanine in the β-alanine auxotroph *E. coli*[20]. These reports indicated that the Uga1 catalysed reaction is reversible, and it may also degrade MSA back to β-alanine, leading to energy waste and decreasing 3-HP yield (Fig. 6a). We therefore deleted *UGA1* in the strain and observed a slight increase in 3-HP production to 10.37 g/L in QLW65 (Fig. 6b). We also deleted alanine transaminases Alt1 and Alt2 that convert pyruvate to alanine, respectively, but it had no impact on the production of 3-HP (Fig. 6b, QLW66 and QLW67). The deletion of *UGA2* to prevent the potential degradation of MSA to malonate did not improve the production of 3-HP, either (Fig. 6b, QLW68). We believe that the *UGA1* deletion only slightly increased 3-HP production because the generation of MSA was restricted. Therefore, we used the strong promoters *CCW12* and *TDH3* to express the MCR-C domain protein (Fig. 6b). The strain with a strong *CCW12* promoter could increase 3-HP production to 10.77 g/L (Fig. 6b, QLW69), indicating that MSA production had been improved. We further deleted *UGA1* in this QLW69 strain and indeed increased 3-HP production to 11.25 g/L (Fig. 6b, QLW71). Meanwhile, we have tested the intracellular concentration of the proposed intermediate β-alanine, and indeed the deletion of *UGA1* in the

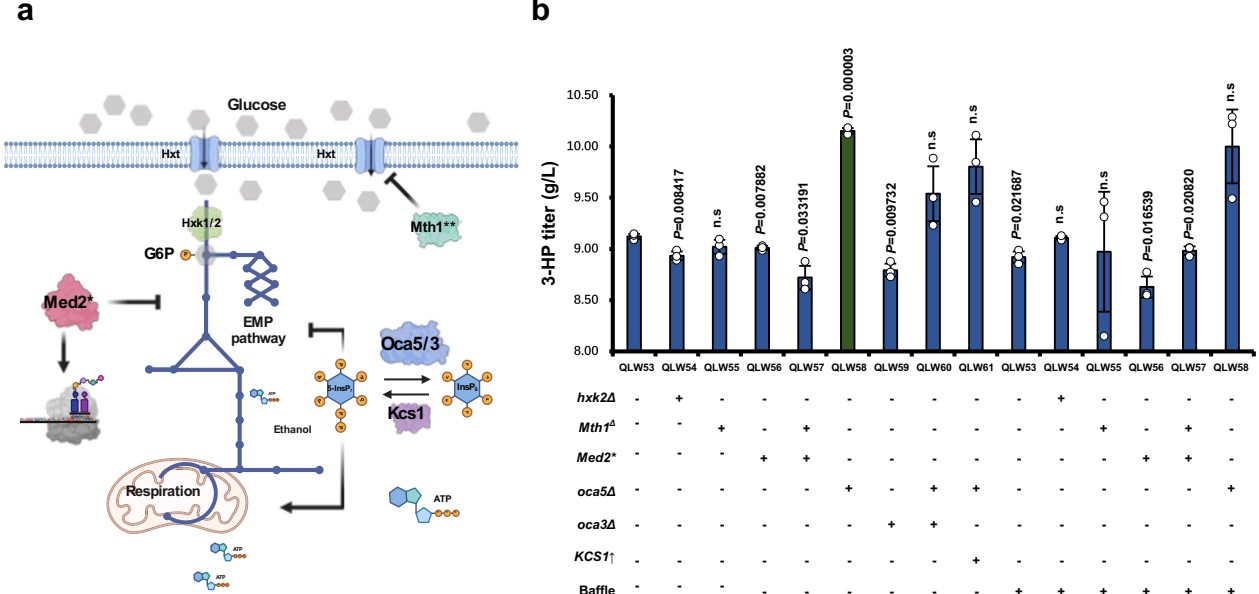

**Fig. 5 | Utilization of inositol pyrophosphate signalling to provide more ATP for 3-HP production. a** Schematic representation of regulation mechanisms involved in glycolysis and respiration in yeast. **b** *OCA5* deletion improved the production of 3-HP to 10.15 g/L. The defined minimal medium with 0.15 g $CaCO_3$ for each flask was used for the 3-HP production. Abbreviations were defined in Supplementary Data 5

and 6. All data were presented as mean ± SD of biological triplicates. Significant comparisons of two groups were indicated in the graphs statistical analysis performed using a two-tailed unpaired Student's *t*-test (*$p < 0.05$, **$p < 0.01$, ***$p < 0.001$). Source data are provided as a Source Data file.

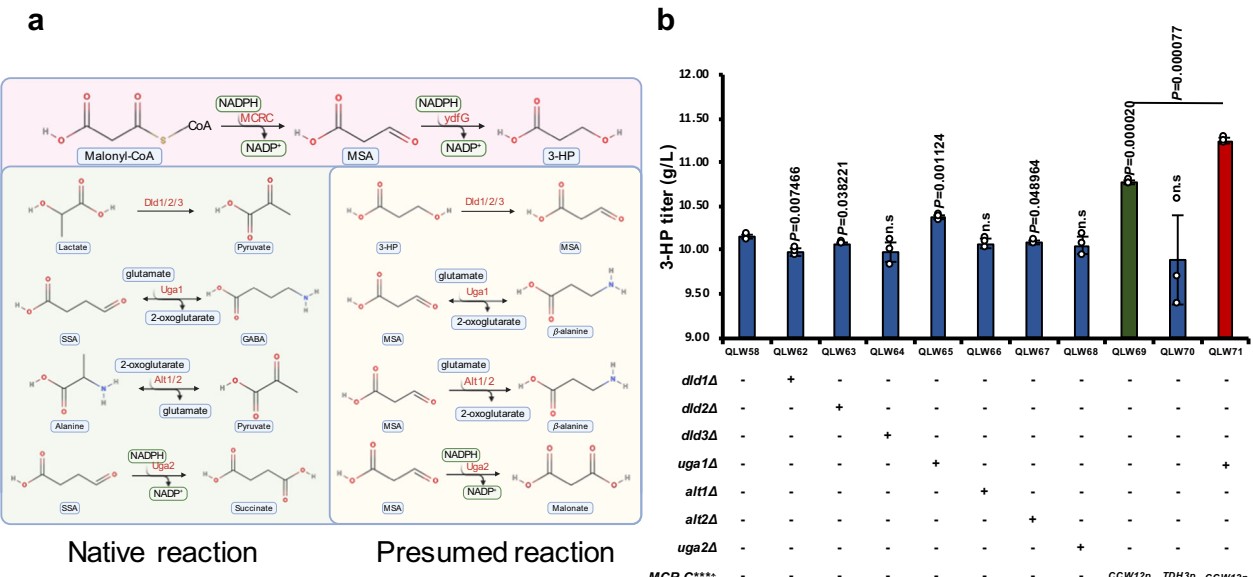

**Fig. 6 | Abolishing potential degradations of MSA improved 3-HP production. a** Candidate enzymes that could potentially degrade 3-HP. **b** Deletion of *UGA1* combined with overexpression of the MCR-C domain using *CCW12* promoter could improve the 3-HP production to 11.25 g/L. Abbreviations were defined in Supplementary Data 5 and 6. The defined minimal medium with 0.15 g $CaCO_3$ for each flask

was used for the 3-HP production. All data were presented as mean ± SD of biological triplicates. Significant comparisons of two groups were indicated in the graphs statistical analysis performed using a two-tailed unpaired Student's *t*-test (*$p < 0.05$, **$p < 0.01$, ***$p < 0.001$). Source data are provided as a Source Data file.

QLW71 strain almost abolished the β-alanine generation (Supplementary Fig. 10). This result indicated the reverse activity of Uga1 converting MSA back to β-alanine. It demonstrated the potential degradation of 3-HP intermediate MSA by the native enzyme in yeast. Our results also underscored the importance that when introducing exogenous metabolite production, the substrate promiscuity and bidirectional crosstalk between the target product and the native metabolic network needs to be taken into consideration.

## Production of 3-HP by the engineered strain enabled efficient conversion to acrylate

By implementing carbon conservation and energy metabolism strategies, we were able to continuously enhance the carbon yield of both 3-HP and biomass, exceeding yeast limit for 3-HP production in terms of bicarbonate accessible under both respiration and fermentation conditions, close to the theoretical yield with concurrent biomass formation with extra bicarbonate providing by $CaCO_3$ (Figs. 1a and 7a).

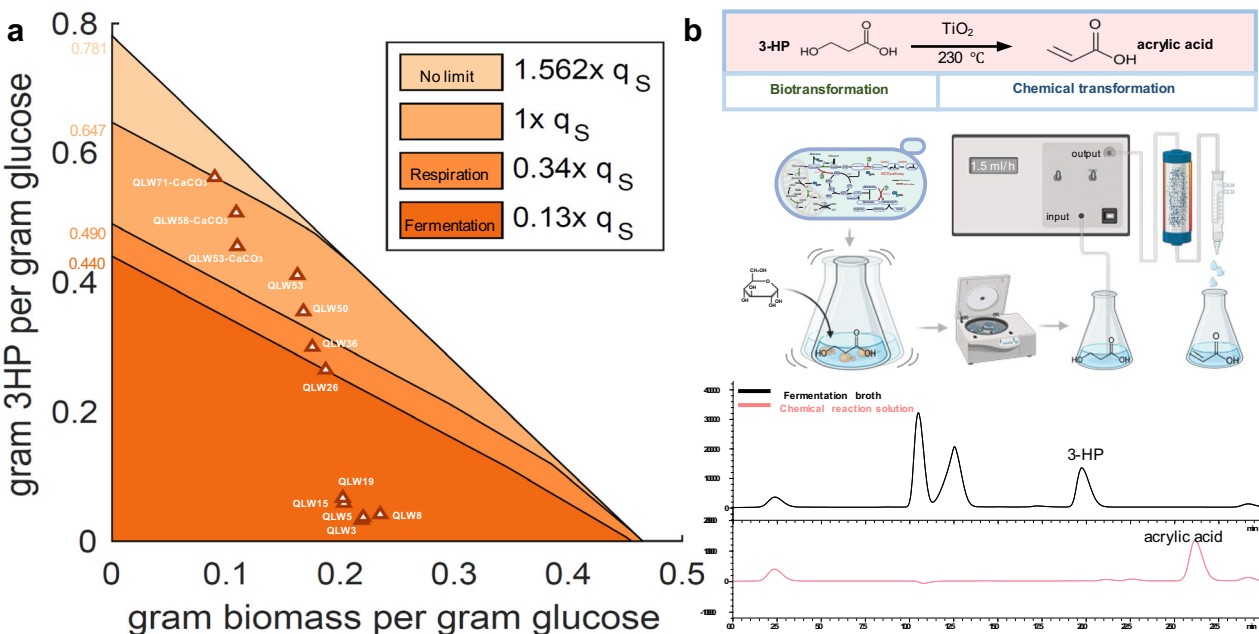

**Fig. 7 | The final 3-HP production and its conversion to acrylic acid. a** The production envelope incorporated with key 3-HP production results reported in this study. Collectively, four of our constructed strains exceeded the 3-HP production boundary with native metabolisms, with QLW71 approaching the inaccessible one to one ratio of bicarbonate influx versus glucose uptake. Qs was substrate (glucose) consumption rate. **b** After a simple centrifugation step to remove cell pellets, the medium could be used directly for dehydration to generated acrylic acid. Source data are provided as a Source Data file.

Meanwhile, the production envelope analysis also revealed a carbon flux competition between the 3-HP production and the biomass generation. With the superposition of multiple strategies, the available carbon resources were insufficient to simultaneously accumulate high levels of both 3-HP and biomass. For example, when further rewiring the carbon flux towards 3-HP from QLW58 to QLW71, the strain could only move to the top left along the production envelope curve and result in less biomass generation (Fig. 7a). Importantly, the yield of QLW71 had reached the production envelope boundary of the bicarbonate supplementation equal to the glucose supplementation.

Through this significant rewiring of carbon flux, we achieved a high 3-HP yield. This high 3-HP titre significantly facilitated the downstream processing, that after simple centrifugation to remove cell pellets, the medium could be directly utilized for acrylate generation. Indeed, the 3-HP present in the medium underwent facile dehydration, facilitated by $TiO_2$ as a catalyst, ultimately yielding acrylate with a remarkable transformation ratio of 100% (Fig. 7b).

## Discussion

Here we employed the production envelope analysis to demonstrate the crucial role of a high flux of bicarbonate in the production of the platform chemical 3-HP. We identified that the limited availability of bicarbonate has been a major obstacle in previous attempts to produce 3-HP (Fig. 1b), while our strategies of thorough analysis and rewiring of the native bicarbonate metabolism combined with external bicarbonate supply, enabled us to surpass this bicarbonate limitation for 3-HP production. Furthermore, to meet the energy demand for bicarbonate utilization, we employed the 5-InsP7 mechanism to enhance the ATP generation yield. In addition, we implemented methods to reduce energy consumption, such as identifying an exporter for 3-HP to minimize energy waste during the 3-HP fermentation process. Moreover, we disrupted the degradation pathway of the intermediate metabolite MSA to prevent the occurrence of energy futile cycles. Collectively, we were able to achieve the in-flux of bicarbonate for 3-HP equivalent to that of glucose. Ultimately, in the QLW71 strain, the yield of 3-HP to substrate reached 0.5625 g/g, which corresponds to 89.3% of the theoretical yield with biomass generation. This

achievement surpassed the production envelope boundary of bicarbonate (Fig. 7a).

The pursuit of a circular and sustainable carbon economy is a key driver for development of biofuel and biobased chemical production from sustainable carbon sources. However, the net impact of biofuel use on global $CO_2$ emissions is still under debate[84,85], leading to continued iterative development[86]. Carbon negativity strategies, such as capturing $CO_2$ or one-carbon feedstock into biomass or target molecules directly, have been proposed but require large energy inputs and may still result in $CO_2$ emissions[87–89]. Our research focused on the high carbon yield production of 3-HP using a multi-strategy approach. Biobased commodity bulk chemicals require high feedstock and carbon yields to achieve economic efficacy and compete with petrochemical-based industries[90]. The pathway for the production of these bulk chemicals must also decrease $CO_2$ release during glucose oxidation to have commercial potential. Our research provides a step towards the development of sustainable and circular carbon economies for biofuel and biobased chemical productions.

## Methods

### Strains, chemicals, and genetic manipulation

An *S. cerevisiae* strain CEN.PK 113-11 C (*MATa SUC2 MAL2-8$^c$ his3Δ1 ura3-52*)[59] was used as the experimental model in this study. All the derivatives strains and their genotypes used in this study were listed in Supplementary Data 1 and 2. For the production of 3-HP, the defined minimal medium[60] was used. In detail, the medium consisted of 7.5 g/L $(NH_4)_2SO_4$, 14.4 g/L $KH_2PO_4$, and 0.5 g/L $MgSO_4·7H_2O$ (the pH of the medium was adjusted to 6.5 with KOH). Trace metal and vitamin solutions were supplemented after the medium was autoclaved at 121 °C for 20 min. Histidine (60 mg/L) and uracil (60 mg/L) were supplemented based on the auxotroph of the plasmid. The calcium carbonate and the sodium bicarbonate were added if needed. To test the dry cell weight (DCW) after the fermentation, the 14.4 g/L $KH_2PO_4$ in the medium with the calcium carbonate fermentation was changed to 3.0 g/L $KH_2PO_4$ same as the medium reported for bioreactor cultivations[91]. For the medium with MES buffer, 100 mM MES monohydrate was added to the defined minimal medium till the pH was

adjusted to 6.5. The carbon source in all the media was 20 g/L glucose. The 3-HP standard was purchased from TCI (TCI, H0297). NaH$^{13}$CO$_3$ was purchased from Sigma-Aldrich (Sigma-Aldrich, 372382), 2-(N-morpholino) ethane sulfonic acid (MES) monohydrate was purchased from Sigma-Aldrich (Sigma-Aldrich, 69892), Nile red was purchased from (Thermo Fisher, N1142). Other chemicals including analytical standards were purchased from Sinopharm Chemical (Beijing, China). All genome modifications were carried out using the GTR-CRISPR system[61]. All plasmids were constructed by Golden Gate assembly. The MCR used in this study was from *Chloroflexus aurantiacus*[21], phosphoketolase with substrate specificity to xylulose-5-phosphate was from *Leuconostoc mesenteroides*[21], and phosphotransacetylase was from *Clostridium kluyveri*[22]. The gene of serine dehydrogenase Ora1 was amplified from the genome of the WT *S. cerevisiae* strain. The NADP$^+$-dependent dehydrogenase YdfG was amplified from the genome of *E. coli* MG 1655. All genes were codon-optimized to *S. cerevisiae* and synthesized by Sangon Biotech (Shanghai, China) if not specified. Details of strains, and DNA fragments were summarized in Supplementary Data 3 and 4, respectively.

### Growth analysis and shake flask fermentation
The strain used for 3-HP production was precultured in the defined minimal medium, then inoculated into a 100 mL shake flask containing 20 mL of the defined minimal medium with a final optical density at 600 nm (OD$_{600}$) of 0.1. The shake flask fermentation was carried out at 30 °C and 220 rpm for 72 h for 3-HP production. The samples' DCW was measured as reported[74]. The biomass composition of CH$_{1.8}$O$_{0.5}$N$_{0.2}$ was assumed[65].

### Analytical methods
Concentrations of 3-HP, glucose, and extracellular metabolites were determined using high-performance liquid chromatography (HPLC, Shimadzu, Japan) with an Aminex HPX-87H column (Bio-Rad, Hercules, USA) at 65 °C, using 0.5 mM H$_2$SO$_4$ as the mobile phase at 0.4 mL/min flow rate for 36 min[17]. Glucose, glycerol, and acrylic acid concentration was detected using a RI-101 Refractive Index Detector, while 3-HP, OAA, acetate, and other extracellular metabolites were detected using a DAD-3000 Diode Array Detector at 210 nm.

### 3-HP tolerance test and intracellular 3-HP measurement
The minimal medium with 50 g/L 3-HP was used. 159 mL 0.5 M citrate solution and 41 mL 1 M Na$_2$HPO$_4$ were used to buffer the medium pH to 3.5. The pH of 3-HP standard liquid was buffered to neutral by NaOH and filtered to sterilize before added into the medium. For the growth test, the Growth Profiler 960 (EnzyScreen) was used. The initial inoculation OD$_{600}$ was 0.01 and 200 µL of cells were cultivated in 96-well microplates at 30 °C and 220 rpm. The strain was cultured in a shake flask with the defined minimal medium and sampled at the exponential phase (OD$_{600}$ around 1). Totally 40 OD$_{600}$ cells were collected and resuspended into 2 mL test medium as an OD$_{600}$ of 20. The test medium was same as the high 3-HP concentration growth test medium in Growth Profiler (pH = 3.5 and 50 g/L 3-HP was added). After be incubated at 30 °C and 220 rpm for 24 h, the culture was centrifuged for 2 min at 3000 × g and washed three times using ultra-pure water. The cells were then lysed by FastPrep-24 5 G for 20 s (6.0 m/s) with glass beads. After centrifuging for 1 min at 12,000 × g, the supernatants, containing the cellular contents released upon lysis, were analysed by HPLC[92].

### Determination of β-alaine
After 72 h of fermentation in the shake flask with the defined minimal medium, 2 mL cell culture was centrifuged for 2 min at 3000 × g to collect the cells. The cell pellet was washed three times with sterile water, re-suspended in 1 mL sterile water and measured the cell OD$_{600}$. 400 µL glass beads were added to this cell suspension to get the cell

wall lysed by FastPrep-24 5 G for 20 s (6.0 m/s). The cell lysis was then centrifuged for 1 min at 12,000 × g and 10 µL supernatant was transferred into a glass HPLC vial for derivatization. Briefly, the AccQ•Fluor Kit (Waters, USA) was used for derivatization. 1 mL of acetonitrile was added to the AccQTag Ultra powder, mixed well, and heated to 55 °C to help it dissolve (heating time should not exceed 15 min). Then, 70 µL of borate buffer was add to the glass HPLC vial containing 10 µL of the cell lysis solution, as well as 20 µL of prepared AccQTag Ultra derivatization solution, mix well, and heated at 55 °C for 10 min. Finally, the heated sample was diluted with derivatization diluent at a ratio of 1:100. The diluted sample was then analyzed using UHPLC-MS/MS. The analysis was performed using Acquity UPLC binary solvent manager, autosampler manager, and column manager (Waters, Milford, MA, USA) coupled with Xevo TQ-S tandem quadrupole mass spectrometer (Waters, Wilmslow, U.K.). Reverse-phase gradient chromatography was employed using an HSS T3 2.1 × 150 mm, 1.8 µm column. The mobile phase A was 0.1% formic acid in water (v/v), while mobile phase B was 1% formic acid in acetonitrile (v/v). The column temperature was maintained at 45 °C. The elution was performed at a flow rate of 0.6 mL/min with a linear gradient: started at 4% B for 0.5 min, increased to 10% B at 2 min, then 28% B at 2.5 min, finally to 95% B for 1 min, and then returned to 4% B for column re-equilibration[93].

### Gene expression analysis
Samples for quantitative real-time PCR (qRT-PCR) analysis were taken when the OD$_{600}$ of the cells reached around 1. The cell pellet was collected by centrifugation at −20 °C, quenched by liquid nitrogen, and stored at −80 °C before use. The RNeasy Mini Kit (QIAGEN, USA) was used for the extraction of total RNA as recommended. The QuantiTect Reverse Transcription Kit (QIAGEN, USA) was used for reverse transcription. 2 µL of produced cDNA was used as the template of the qPCR reaction with the DyNAmo Flash SYBR Green qPCR Kit (Thermo Scientific, USA). Quantitative RT-PCR was performed on Stratagene Mx3005P (Agilent Technologies, USA). The housekeeping gene *ACT1* was selected as the reference gene. All the primers were listed in Supplementary Data 4.

### Catalytic dehydration of 3-HP to acrylic acid
Acrylic acid synthesis was carried out in a fixed-bed tubular reactor (i.d. = 8 mm, o.d. = 12 mm, length = 30 cm). After culturing the strains for 72 h in a defined minimal medium, the fermentation broth was sterilized by filtration and diluted 6 times with ultrapure water. And then 3 g TiO$_2$ as the catalyst (20–40 mesh) was loaded into the reactor tube between two layers of quartz wool and preheated at 230 °C in the furnace. The diluted fermentation broth was delivered using a metering pump with a rate of 100 mL/h for 5 min, and then with a constant rate of 1.5 mL/h. The eluate containing acrylic acid was condensed in a cold-water bath. Liquid samples of the first 30 min were discarded. In addition, 4 samples were collected for each measurement.

### Genome-scale metabolic modelling
The genome-scale metabolic model of *S. cerevisiae* Yeast8[17], was expanded by adding two heterologous reactions, i.e., malonyl-CoA reduction and MSA reduction, as well as the secretion of 3-HP. The resulting model with the default constraints was used to estimate the yield of 3-HP on glucose. To do so, the uptake rate of glucose was fixed at 1 mmol/gDCW/h, the growth rate was adjusted, and the secretion of 3-HP was maximized for each growth rate. To simulate the effect of native bicarbonate production on the yield, the upper bound on the rate of the bicarbonate formation in the model, which converts carbon dioxide to bicarbonate, was set at different values normalized to the glucose uptake rate. Model simulations were performed using the COBRA toolbox[94] in MATLAB with the CPLEX as an optimization solver.

## Phylogenetic and structure analysis

Amino acid sequences from the SLC4 family (InterPro ID: IPR003020) and SLC26 family (InterPro ID: IPR001902) were aligned and trimmed using NGPhylogeny.fr[95]. Only key species (*S. cerevisiae*, *Arabidopsis thaliana*, and *Homo sapiens* for SLC4 and *S. cerevisiae*, *E. coli*, *A. thaliana*, and *H. sapiens* for SLC26) were included. MEGA11 was used to construct a maximum likelihood phylogenetic tree and tested with 500 bootstraps[96]. iTOL v6 was used in the modification of the phylogenetic tree[97]. The structure of human SLC26A11 (Uniprot ID Q86WA9), yeast Sul1 (Uniprot ID P38359), and yeast Sul2 (Uniprot ID Q12325) were obtained from AlphaFoldDB (https://alphafold.ebi.ac.uk/)[98]. FATCAT 2.0 (https://fatcat.godziklab.org/)[99] was used to compare the structures of SLC26A11 with Sul1 and Sul2.

## Isotope $^{13}C$ labelling analysis

The minimal medium with 30 mM $NaH^{13}CO_3$ (Sigma-Aldrich) in a shake flask was used for the isotope labelling experiment. Strains were inoculated with $OD_{600}$ 0.01 and sampled after 72 h. 2 mL medium was transferred immediately into the 15 mL tube with 8 mL pure methanol (precooling in −80 °C fridge) and vortexed around 1 s for quenching samples. Then, the mix was centrifuged at $5000 \times g$ for 3 min at 4 °C. Remove the supernatant and resuspend the precipitate with 3 mL of precooling methanol, $5000 \times g$ centrifuged for 3 min, remove the supernatant, keep on ice. For hydrolysis of cellular components, 6 mL 6 N hydrochloric acid was added to quenched cells and transferred to 10 mL glass digestion tube, cells were hydrolysed for 16–20 h in metal bath at 115 °C and dried in a heating block at 85 °C, then 600 μL of ultrapure water was added, and the sample was centrifuged at the maximum speed for 1 min[100]. Before the MS analysis, the sample was filtrated with a 0.22 μm pore size filter membrane. The isotopic labels of metabolites were determined with UPLC-MS/MS (SCIEX TRIPLETOF 6600).

## Nile red staining

After 96 h of cultivation in the defined minimal medium, 250 μL cell culture was transferred into a sterile 1.5 mL tube. Then 25 μL freshly prepared DMSO: PBS (1:1) buffer and 25 μL Nile red (60 μg/mL in acetone) were added. The working concentration of Nile red was 5 μg/mL. The sample was well mixed and then incubated away from the light for 5 min at room temperature. The strain was washed twice with 300 μL PBS buffer and resuspended in 300 μL PBS buffer. 5 μL of the cell sample was pipetted onto a glass microscope slide, placed with the cover slip firmly, and observed using the green fluorescence channel with the excitation wavelength at 488 nm and emission wavelength at 509 nm. For quantitative analysis, the yeast culture in the defined minimal medium was harvested at 96 h, centrifuged and resuspended twice with PBS buffer to achieve an $OD_{600}$ of 2.0. Transfer 250 μL of each yeast sample to the corresponding wells of a 96-well plate, ensuring proper mixing as yeast cells tend to settle. 12.5 μL DMSO: PBS (1:1) buffer for each well was added in advance. 25 μL of the freshly prepared 60 μg/mL Nile red stock solution was added for each well to a final 5 μg/mL concentration. The microplate reader with excitation and emission wavelengths was set at 485 nm and 535 nm, respectively, and a top 50% mirror was used. Optimal gain settings and an orbital shake for 10 s before reading were employed and the plate was read immediately after Nile red addition for the necessary number of cycles[101].

## Reporting summary

Further information on research design is available in the Nature Portfolio Reporting Summary linked to this article.

## Data availability

The data supporting the findings of this work are available within the paper and the Supplementary Information files. A reporting summary for this article is available as a Supplementary Information file. Source data are provided with this paper.

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

## Acknowledgements

This work was supported by National Key Research and Development Program of China (2018YFA0900100), National Natural Science Foundation of China (22078012), Fundamental Research Funds for the Central Universities (buctrc202304), the Beijing Advanced Innovation Centre for Soft Matter Science and Engineering, Beijing University of Chemical Technology, the Novo Nordisk Foundation (NNF20CC0035580) and the Swedish Research Council FORMAS. This paper was edited using ChatGPT. We also thank Professor Jiqin Zhu from Beijing University of Chemical Technology for his kind help on the acrylic acid experiment.

## Author contributions

N.Q., Z.L., Yun C., and J.N. designed the research; N.Q., L.L., X.W., X.J., Yu C., C.L., P.L., Y.Z., W.Y., and J.J. carried out the experiment; N.Q., L.L., J.X., S.S., T.T., J.N., Yun C., and Z.L. analysed the data; N.Q., J.N., Yun. C., and Z.L. wrote the paper; J.N., Yun C., and Z.L. supervised the research. N.Q. and L.L. contributed equally.

## Competing interests

The authors declare no competing interests.
