## [Peer Review File · Nature Communications]

Increased CO₂ Fixation Enables High Carbon-Yield Production of 3-Hydroxypropionic Acid in YeastReviewers' Comments:

Reviewer #1:

Remarks to the Author:

In this research article, the authors successfully synthesized 3-hydroxypropionic acid (3-HP) with a high carbon yield in *Saccharomyces cerevisiae*. They employed metabolic modeling to identify the bicarbonate flux as the limiting factor for 3-HP production via the malonyl-CoA reductase pathway. To enhance CO₂ utilization, several strategies were implemented. These included the introduction of the phosphoketolase pathway and the overexpression of SUL1, which was identified as a critical bicarbonate transporter. Through extensive engineering efforts targeting multiple aspects of the metabolic pathway, both novel and previously established, the authors achieved an impressive 3-HP yield, reaching 89.3% of the theoretical maximum. While this study has made significant contributions, there are some issues that need to be addressed, as outlined below.

1. Line 103, The first strain QLW1 contains MCR-C mutant. However, the details of the MCR-C mutant was not provided until it is described on line 170. Please specify the traits of the MCR-C mutant the first time it is mentioned in the manuscript.
2. Line 106: Please incorporate additional information regarding the phosphoketolase (PK) pathway, particularly elucidating which specific genes were overexpressed as part of this pathway.
3. Line 144-145. Please clarify whether the accumulation of oxaloacetate (OAA) was solely attributable to the overexpression of SUL1 or whether it occurred in conjunction with the overexpression of STB5. In other words, it needs to be specified whether the observed OAA accumulation was also observed in the QLW8 strain overexpressing only SUL1. Additionally, it raises curiosity as to why the authors chose to maintain the overexpression of both STB5 and SUL1 despite the observed adverse effects when these genes were overexpressed in combination.
4. Line 185-186. Please provide a more comprehensive explanation regarding the selection process of Ora1 and ydfG from a pool of proteins containing SDR. The rationale for choosing Ora1 can be found later in the manuscript (lines 323-324).
5. Fig. 2g. The presented data does not provide sufficient evidence to support the claim of reduced lipid production in the QLW36 strain. It should either be omitted or replaced with more quantifiable data for a stronger argument.
6. Line 231, The reference cited as #50 does not contain information about increased lactate production resulting from the deletion of GSF2.
7. Fig. 4d. In this study, CaCO₃ was introduced to augment bicarbonate availability for malonyl-CoA generation. Nevertheless, it's noteworthy that CaCO₃ is extensively employed as a neutralizing agent in lactic acid production processes. Despite the utilization of MES buffering as a control, it remains challenging to differentiate between the effects of neutralization and bicarbonate supplementation. To eliminate the influence of neutralization, please investigate other neutralizing agents used for organic acid production, such as calcium hydroxide that does not provide bicarbonate, and comparing their impacts to those of CaCO₃. Furthermore, it would be beneficial to include cell growth curves when various concentrations of CaCO₃ were treated. Furthermore, on line 417, it is mentioned that "calcium carbonate and sodium bicarbonate were added if needed," but there is no information available about experiments involving sodium bicarbonate. Can it be confirmed whether the addition of sodium bicarbonate also resulted in increased 3-HP production?
8. Please provide details regarding the culture conditions for Fig. 5b and Fig. 6b. Are the media used in these figures contain 75 mM CaCO₃, as shown in Fig. 4d? If this is the case, it is recommended to label QLW58 and QLW71 in a similar manner as QLW53-CaCO₃ in Fig. 7a to enhance clarity. Also, it should be mentioned when describing the final 3-HP yield of the engineered strain.

Reviewer #2:

Remarks to the Author:

The work of Qin et al. is an impressive example of genetic engineering of microbial cells for value-added production. The authors engineered yeast cells to produce 3-hydroxypropionate, one of the top-

12 value-added chemicals for the production from biomass, according to the DOE report. The authors used a combination of very different approaches to approach the maximum theoretical yield of the production. These methods include flux analysis, medium optimization (e.g., bicarbonate supply), enzyme engineering, introduction of new enzymes and pathways, deregulation, overexpression or repression of a number of enzymes involved in the process, identification of bicarbonate transporters, improvement of the 3-hydroxypropionate tolerance of the cells, and limitation of the Crabtree effect. The authors were also able to show that the 3-hydroxypropionate produced can be used for acrylate production. All in all, it is an interesting and informative story, and I will be glad to see it published. However, I have a number of minor comments that should be addressed before publication.

Although the manuscript is generally well written, in some cases it needs language editing (some examples below), and I recommend to show it to a native speaker with knowledge of microbiology.

l. 57: In fact, Ref. 11 describes the 3-hydroxypropionate bi-cycle in *Chloroflexus* and not the HP/HB cycle. The discussion on this cycle should probably be added to the introduction, as malonyl-CoA reductase used in the work comes actually from *Chloroflexus*.

l. 101-104: please also cite the original work identifying malonyl-CoA reductase in *Chloroflexus* (Hügler et al., 2002, doi: 10.1128/JB.184.9.2404-2410.2002).

l. 112: "(oxPP pathway)": "(oxPP) pathway".

l. 153: what is the function of CIT1?

l. 159: what is the function of Idp2?

l. 165: what is AMPK pathway?

l. 184-185: please provide the data that support your claim that the obtained enzyme was "the most efficient malonyl-CoA reductase reported".

l. 186 and elsewhere: ydfG; YdfG?

l. 203-208: I do not see any difference between the control cells and the engineered strain. Please present the results in the quantitative way, as it is currently not convincing.

l. 233-249 and elsewhere in the manuscript: please always provide the strain numbers to help the reader to follow your text by looking the figures, where the strain numbers only are shown. In some cases, this information is missing in the text.

l. 243: please delete "."

l. 248-249 versus l. 239: "significantly improved the production [...] 6.39 g/L" but earlier: "with the production levels reaching 6.45 g/L": 6.39 g/L does not really look like improvement here.

l. 254-255: "To our knowledge, this was the first exporter of 3-HP reported": but the function of Esbp6 has not been really proven here to make this claim.

l. 304: "Deleting HXK2": which enzyme fulfill the function of HXK2 then?..

l. 442-444: how the chemicals were detected? Which detector was used for that? RI detector? PDA detector? (if yes, which wavelength?) MS?

l. 485-486: "method was followed as previously described": the previously described method was used?

l. 488: "then add 600 μ L of ultrapure water, centrifuged at the maximum speed": "then 600 μ L of ultrapure water was added, and the sample was centrifuged at the maximum speed".

l. 491: research: reported?

- I. 516-517: I do not understand this sentence.
- I. 521: reduce: reducing
- I. 521-522: "could [...] enhanced": either "enhanced" or "could [...] enhance".

- I. 532, 533: CO₂ releasing reactionS, bicarbonate utilizing reactionS

- I. 536: "their expression through by chromosome integration"???
- I. 537: please delete "were"
- I. 545: Increase in the production?
- I. 548-549: increase cell tolerance to high concentrations of 3-HP, compared to the WT strain?

- I. 563: isomerized relation???
- I. 567-568: I do not understand the title of this figure.

Reviewer #3:

Remarks to the Author:

This paper describes the high production of 3-hydroxypropionic acid (3-HP), the critical precursor for acrylic acid. The authors introduced and deleted various proteins, including modifying several intracellular pathways, and higher CO₂ fixation, leading to a significant increase in the 3-HP.

This study is highly appreciated since it has succeeded in producing a quantity of 3HP that far surpasses known reports through careful planning and extensive experimentation. However, the following points should be revised or addressed.

Major comments.

1. Some proteins should be characterized as those proteins function as the authors proposed. It is unclear whether SUL1 imports bicarbonate or production increase results from the side effect of SUL1 protein, like sulfate import. Efflux activity of 3-HP by ESBP6 was not also demonstrated. Growth enhancement might be observed by its other effect. Using NaH¹³CO₃ and 3-HP, cellular transport would be possible to characterize those transporters. Also, for UGA1, its biochemical characterization or analysis of metabolic profile alteration by deleting *uga1*, which might show the increase or decrease of the proposed intermediates or other metabolites, would be desirable. Those analyses would strengthen this study as the comprehensive modifications, including substrate import, product efflux, and inhibition of intermediate degradation, benefit the high production of valuable metabolites.

2. As for Nile red staining (Fig. 2g), quantitative data, such as the diameter and the number of oil bodies, should be shown.

Minor comments.

1. L. 136. Reference 32 mainly describes SHST1 from *Stylosanthes hamata*, less statement of *sul1/2*. For *sul1/2*, other papers would be appropriate.

2. L. 149-151. The data that acetate accumulation should be shown as a supplement.

3. L. 158. Please add the reference for *Yhm2*

4. L. 184-188. Please add the references for *Ora1* and *ydfG*.

5. L. 200-202. Please add the reference for *POX1* and *POX2* and explain those functions. Adding peroxisome in Fig. 2e might be desirable if these oxidations occur in the peroxisome. The slight effect of *POX1* and *POX2* might be due to the bottleneck of peroxisomal transport of metabolites.

6. L. 236-239. Please add the reference of SUR1.
7. L. 243. "and" might be "at".
8. L. 442-444. Please re-check the mobile phase; the authors did not use organic solvent for HPLC?
9. Please re-check that the description of Fig. 2c is correct.
MCR-N, malonate semialdehyde reductase; MCR-C, Malonyl-CoA reductase
10. Fig. 2g legend. Please state which strain was used as the control.
11. Fig. S1. Please re-check the description in the figure. ARTp-STB5 and TEF1p-STB5 would be correct. In the legend and figure, an inverse description is shown.

Reviewer #1 (Remarks to the Author):

*In this research article, the authors successfully synthesized 3-hydroxypropionic acid (3-HP) with a high carbon yield in *Saccharomyces cerevisiae*. They employed metabolic modeling to identify the bicarbonate flux as the limiting factor for 3-HP production via the malonyl-CoA reductase pathway. To enhance CO₂ utilization, several strategies were implemented. These included the introduction of the phosphoketolase pathway and the overexpression of *SUL1*, which was identified as a critical bicarbonate transporter. Through extensive engineering efforts targeting multiple aspects of the metabolic pathway, both novel and previously established, the authors achieved an impressive 3-HP yield, reaching 89.3% of the theoretical maximum. While this study has made significant contributions, there are some issues that need to be addressed, as outlined below.*

1. Line 103, The first strain QLW1 contains MCR-C mutant. However, the details of the MCR-C mutant was not provided until it is described on line 170. Please specify the traits of the MCR-C mutant the first time it is mentioned in the manuscript.

Response: We thank the reviewer for noting this. The traits of the mutated version of MCR-C^{N941V K1107W S1115R} was added at the first time when it was mentioned (Line 104).

2. Line 106: Please incorporate additional information regarding the phosphoketolase (PK) pathway, particularly elucidating which specific genes were overexpressed as part of this pathway.

*Response: We have now added the details of the PK pathway including the genes and their origins, such as: "the phosphoketolase (PK) pathway reported in our previous study, including a xylulose-5-phosphate-preferred phosphoketolase (xPK) from *Leuconostoc mesenteroides* and a phosphotransacetylase (PTA), from *Clostridium kluyveri*" (Line 110-112).*

*3. Line 144-145. Please clarify whether the accumulation of oxaloacetate (OAA) was solely attributable to the overexpression of *SUL1* or whether it occurred in conjunction with the overexpression of *STB5*. In other words, it needs to be specified whether the observed OAA accumulation was also observed in the QLW8 strain overexpressing only *SUL1*. Additionally, it raises curiosity as to why the authors chose to maintain the overexpression of both *STB5* and *SUL1* despite the observed adverse effects when these genes were overexpressed in combination.*

Response: We thank the reviewer for pointing this out. The accumulation of oxaloacetate was only observed in the strain with the overexpression of *SUL1* and *STB5*, whereas there was no oxaloacetate accumulation in the *SUL1* overexpression strain QLW8. We have now clarified this in the revised text (Line 155-156). Although negative results were obtained combining *Stb5* with *Sul1*, theoretically it should work. The precursors of 3-HP synthesis include NADPH, acetyl-CoA, bicarbonate and ATP. The *Stb5* strategy could increase the carbon flux into the oxPP pathway and the following PK pathway for NADPH and acetyl-CoA supply, while *Sul1* could potentially increase intracellular bicarbonate concentration. Thus, we chose to trouble shoot, and through additional strategies and further improved 3-HP production to 1.34 g/L, as shown in Fig. 1h.

Fig.1h. Rewiring the carbon flux from OAA to the high production of 3-HP. Overexpressing the phosphatase *Ptc7* could significantly increase the 3-HP production based on overexpressing *CIT1*, introducing *Cit1*^{S462A} mutation, overexpressing the transporter *Yhm2* to transport the OAA into mitochondria, and overexpressing cytosolic NADP-specific isocitrate dehydrogenase *Idp2*.

4. Line 185-186. Please provide a more comprehensive explanation regarding the selection process of *Ora1* and *ydfG* from a pool of proteins containing SDR. The rationale for choosing *Ora1* can be found later in the manuscript (lines 323-324).

Response: We thank the review for this suggestion. The rationale of *Ora1* and *YdfG* was added as: "To find a more efficient domain converting MSA to 3-HP, we performed domain swapping using SDRs from different species and obtained the improved malonyl-CoA reductase. Specially, we identified potential MCR-N-SDR1 domains through the substrate structural similarity, such as *Ora1* (reported as the serine dehydrogenase) from *S. cerevisiae*, and

YdfG (reported as the NADP⁺-dependent dehydrogenase with broad substrate specificity on 3-hydroxy acids) from *E. coli* (Fig. 2c). Serine can be seen as 2-amino-3-hydroxypropionic acid, which could be regarded as the amino group substituted at the second carbon atom site of 3-HP. The substrate structural similarity together with the broad substrate specificity to 3-hydroxy acids of Ora1 and YdfG indicated the potential of these enzymes for 3-HP production.” (Line 192-200).

Fig.2c. Homologous modelling structure of the domains in malonate semialdehyde reductase and malonyl-CoA reductase from *C. aurantiacus*.

5. Fig. 2g. The presented data does not provide sufficient evidence to support the claim of reduced lipid production in the QLW36 strain. It should either be omitted or replaced with more quantifiable data for a stronger argument.

Response: We thank the reviewer for the suggestion. We have now added the Nile red based quantification data in Fig.2h. The new results showed that the relative fluorescence unit of Nile red (stain of lipid droplet) in strain QLW36 with up-regulated β -oxidation pathway and down-regulated fatty acid biosynthesis pathway was significantly decreased compared with that of the control strain QLW26.

Fig.2h. The fluorescence intensity of neutral lipids stained with Nile red in QLW26 and QLW36 was quantified in Relative Fluorescence Units (RFU).

6. Line 231, The reference cited as #50 does not contain information about increased lactate production resulting from the deletion of GSF2.

Response: We are so sorry for this mis-citations. The correct reference about increased lactate production resulting from the deletion of GSF2 has now been added (Line 244).

7. Fig. 4d. In this study, CaCO₃ was introduced to augment bicarbonate availability for malonyl-CoA generation. Nevertheless, it's noteworthy that CaCO₃ is extensively employed as a neutralizing agent in lactic acid production processes. Despite the utilization of MES buffering as a control, it remains challenging to differentiate between the effects of neutralization and bicarbonate supplementation. To eliminate the influence of neutralization, please investigate other neutralizing agents used for organic acid production, such as calcium hydroxide that does not provide bicarbonate, and comparing their impacts to those of CaCO₃. Furthermore, it would be beneficial to include cell growth curves when various concentrations of CaCO₃ were treated. Furthermore, on line 417, it is mentioned that "calcium carbonate and sodium bicarbonate were added if needed," but there is no information available about experiments involving sodium bicarbonate. Can it be confirmed whether the addition of sodium bicarbonate also resulted in increased 3-HP production?

Response: We thank the reviewer for this comment. CaCO₃ has been used as a neutralizing agent in lactic acid production processes. In such a process it does not involve the large requirement of bicarbonate like 3-HP, hence CaCO₃ is mainly function as pH buffering. To verify if the contribution of CaCO₃ is solely caused by the influence of neutralization, we tested 3-HP production using different amounts of calcium hydroxide to buffer the pH of medium as suggested. As shown in Fig. S9, when adding up to 40 mM calcium hydroxide (0.0592 g/flask), the titer of 3-HP only had a slight increase. While adding more calcium hydroxide, the medium became suspension and the cells stopped growth (data not shown). Thus, together with the utilization of MES buffering as a control, the effect of CaCO₃ could not be solely explained by the influence of neutralization, but rather a combination effect of both neutralization effect together with a more important augment of bicarbonate availability.

Figure S9. Effects of calcium hydroxide on 3-HP production. The impact of varying calcium hydroxide concentrations added to the medium on 3-HP production was investigated. 0 mM, 10 mM, 20 mM, and 40 mM represent 0 g /flask, 0.0148 g/flask, 0.0296 g/flask, and 0.0592 g/flask. All data points were depicted as the mean \pm standard deviation (SD) derived from biological triplicates.

As for cell growth curves when various concentrations of CaCO₃ were treated, we selected conditions with or without the optimal 0.15 g/flask calcium carbonate. As shown in Fig. S8, cell growth was decreased in strain QLW53 when added with CaCO₃, compared with that without the supplementation of CaCO₃.

Figure S8. The effects of CaCO₃ on the QLW53 strain. 75 mM (0.15 g/flask) CaCO₃ was added to test the growth. The genotype is listed in Table S2. All data were presented as mean \pm SD of biological triplicates.

Meanwhile, we are sorry for not clarifying the medium composition with different kinds of carbonates, that have now been added in the figure legends. Briefly, through calculation (Fig. 1a) we believed that the intracellular bicarbonate availability could be the key factor to further increase 3-HP production. So, we had directly supplemented sodium bicarbonate in our first strain evaluated (QLW1, Fig. 1c), till Fig. 4d that calcium carbonate was first employed. Upon your suggestion, we noticed that we better add an experiment comparing the 3-HP production in the QLW1 strain with or without sodium bicarbonate. We have now added this data in Fig. 1c as well, that supported our hypothesis that the addition of bicarbonate indeed improved 3-HP production.

Fig.1 (a) The production envelope analysis based on the Yeast8 model with the malonyl-CoA reductase reaction for 3-HP production. **(c)** Adjusting *STB5* expression by *ARTp* could together with the expression optimization of MCR domains, *Acc1* and the PK pathway enhance 3-HP production to 0.74 g/L.

8. Please provide details regarding the culture conditions for Fig. 5b and Fig. 6b. Are the media used in these figures contain 75 mM CaCO₃, as shown in Fig. 4d? If this is the case, it is recommended to label QLW58 and QLW71 in a similar manner as QLW53-CaCO₃ in Fig. 7a to enhance clarity. Also, it should be mentioned when describing the final 3-HP yield of the engineered strain.

Response: We thank the reviewer for pointing this out. In both Fig. 5b and Fig. 6b, the medium contained 75 mM CaCO₃ (0.15 g/flask), as shown in Fig. 4d. We have now adjusted the labelling of QLW58 and QLW71 in a manner consistent with QLW53-CaCO₃ in Fig. 7a. We have now also refined our description when describing the final 3-HP yield of the engineered strain (Line 376).

Fig. 7a. The production envelope incorporated with key 3-HP production results reported in this study.

Reviewer #2 (Remarks to the Author):

The work of Qin et al. is an impressive example of genetic engineering of microbial cells for value-added production. The authors engineered yeast cells to produce 3-hydroxypropionate, one of the top-12 value-added chemicals for the production from biomass, according to the DOE report. The authors used a combination of very different approaches to approach the maximum theoretical yield of the production. These methods include flux analysis, medium optimization (e.g., bicarbonate supply), enzyme engineering, introduction of new enzymes and pathways, deregulation, overexpression or repression of a number of enzymes involved in the process, identification of bicarbonate transporters, improvement of the 3-hydroxypropionate tolerance of the cells, and limitation of the Crabtree effect. The authors were also able to show that the 3-hydroxypropionate produced can be used for acrylate production. All in all, it is an interesting and informative story, and I will be glad to see it published. However, I have a number of minor comments that should be addressed before publication.

Although the manuscript is generally well written, in some cases it needs language editing (some examples below), and I recommend to show it to a native speaker with knowledge of microbiology.

Response: We thank the reviewer for this suggestion. We have now carefully revised the manuscript accordingly.

I. 57: In fact, Ref. 11 describes the 3-hydroxypropionate bi-cycle in *Chloroflexus* and not the HP/HB cycle. The discussion on this cycle should probably be added to the introduction, as malonyl-CoA reductase used in the work comes

actually from *Chloroflexus*.

Response: We are sorry for the typo on this. The right reference of 3-hydroxypropionate/4-hydroxybutyrate (HP/HB) cycle has now been added, so as the reference and the description about the 3-hydroxypropionate bi-cycle (Line 58).

l. 101-104: please also cite the original work identifying malonyl-CoA reductase in Chloroflexus (Hügler et al., 2002, doi: 10.1128/JB.184.9.2404-2410.2002).

Response: We have now added the suggested reference in line 103.

l. 112: "(oxPP pathway)": "(oxPP) pathway".

Response: Fixed.

l. 153: what is the function of CIT1?

Response: We are sorry for the confusion. The function of *CIT1* has now been added as: "*CIT1* encoding the mitochondrial citrate synthase that converts OAA and acetyl-CoA to citrate" (Line 161-162).

l. 159: what is the function of ldp2?

Response: We are sorry for the confusion. The function of *ldp2* has now been added as: "the cytosolic NADP-specific isocitrate dehydrogenase *ldp2*" (Line 168-169).

l. 165: what is AMPK pathway?

Response: We are sorry for the confusion. The AMPK pathway is the AMP-activated kinase pathway. We have now added the full name as: "the AMP-activated kinase (AMPK) signalling pathway" (Line 175-176).

l. 184-185: please provide the data that support your claim that the obtained enzyme was "the most efficient malonyl-CoA reductase reported".

Response: We are sorry for this demonstration. Since our data was just interpreted based on production of 3-HP, we have revised the description as: "the improved malonyl-CoA reductase" (Line 193-194).

l. 186 and elsewhere: ydfG: YdfG?

Response: Fixed.

l. 203-208: I do not see any difference between the control cells and the engineered strain. Please present the results in the quantitative way, as it is currently not convincing.

Response: We thank the reviewer for the suggestion. We have now added the Nile red quantification data in Fig. 2h. The new results showed that the relative fluorescence unit of Nile red (stain of lipid droplet) in strain QLW36 with up-regulated β -oxidation pathway and downregulated fatty acid biosynthesis pathway was significantly decreased compared with that of the control strain QLW26.

Fig. 2h. The fluorescence intensity of neutral lipids stained with Nile red in QLW26 and QLW36 was quantified in Relative Fluorescence Units (RFU).

l. 233-249 and elsewhere in the manuscript: please always provide the strain numbers to help the reader to follow your text by looking the figures, where the strain numbers only are shown. In some cases, this information is missing in the text.

Response: We are sorry that we did not present more clearly in the original version of the manuscript and thanks for the suggestion. We have now added the strain numbers in line 251-252 and also throughout the paper.

l. 243: please delete “.”

Response: Fixed.

l. 248-249 versus l. 239: “significantly improved the production [...] 6.39 g/L” but earlier: “with the production levels reaching 6.45 g/L”: 6.39 g/L does not really look like improvement here.

Response: We are sorry for the confusion. All the results in line 252-253, and line 262-263 were assessed in comparison with the QLW36 strain as a reference (Fig. 3b). We have now revised the text to make this clear.

Fig. 3b. Overexpression of *ESBP6*, deletion of *SAM2*, *GSF2*, or *ERF2* could significantly improve the production of 3-HP.

I. 254-255: “To our knowledge, this was the first exporter of 3-HP reported”: but the function of *Esbp6* has not been really proven here to make this claim.

Response: We totally agree with the reviewer and thank the reviewer for the suggestion. We have now added the measurement of the intracellular 3-HP concentration in both the *ESBP6* overexpression strain and the wild-type (WT) strain (Fig. S6). The overexpression of *ESBP6* led to a significant decrease in intracellular 3-HP content compared with that of the WT strain. These results indicated that *Esbp6* could help exporting 3-HP out of the cell, thereby increasing the tolerance of yeast and improving 3-HP production. Meanwhile, we have also weakened the expression in the main text, as “To our knowledge, this could be the first evidence for potential 3-HP exporters.” (Line 269-270).

Figure S6. Effects of *ESBP6* gene on intracellular 3-HP concentration. The intracellular 3-HP content was measured after the treatment of strains with 50 g/L 3-HP for 24 hours to assess the impact of the *ESBP6* gene. All data points were represented as the mean \pm standard deviation (SD) of biological triplicates.

I. 304: "Deleting HXK2": which enzyme fulfill the function of HXK2 then?..

Response: We are sorry that we did not present more clearly in the original version of the manuscript and thanks for the suggestion. The description of *HXK2* has now been changed to: "When glucose is transported inside the cell, it is first phosphorylated primarily by Hxk2, and also by Glk1, Hxk1" (Line 323-324).

I. 442-444: how the chemicals were detected? Which detector was used for that? RI detector? PDA detector? (if yes, which wavelength?) MS?

Response: We are sorry for the confusion. The detector that was used for chemicals has now been added as: "Glucose, glycerol, and acrylic acid concentration was detected using a RI-101 Refractive Index Detector, while 3-HP, OAA, acetate, and other extracellular metabolites were detected using a DAD-3000 Diode Array Detector at 210 nm." (Line 468-471).

I. 485-486: "method was followed as previously described": the previously described method was used?

Response: Fixed.

I. 488: "then add 600 μ L of ultrapure water, centrifuged at the maximum speed": "then 600 μ L of ultrapure water was added, and the sample was centrifuged at the maximum speed".

Response: Fixed

I. 491: research: reported?

Response: Fixed.

I. 516-517: I do not understand this sentence.

Response: We are sorry that we did not present this more clearly in the original version of the manuscript. We have revised the text to: "The DCW was either measured directly or estimated based on OD₆₀₀ (DCW equals to 70% of OD₆₀₀)." (Line 595-596).

l. 521: reduce: reducing

Response: Fixed.

l. 521-522: “could [...] enhanced”: either “enhanced” or “could [...] enhance”.

Response: Fixed.

l. 532, 533: CO2 releasing reactionS, bicarbonate utilizing reactionS

Response: Fixed.

l. 536: “their expression through by chromosome integration”???

Response: We are sorry that we did not present this clearly in the original version of the manuscript. We have now revised the text as: “Genes encoding split MCR enzymes were either integrated in the chromosome or expressed using the plasmid system.” (Line 610-611).

l. 537: please delete “were”

Response: Fixed.

l. 545: Increase in the production?

Response: Fixed.

l. 548-549: increase cell tolerance to high concentrations of 3-HP, compared to the WT strain?

Response: We are sorry for not clarifying this, and thanks for the suggestion. We have revised our description as “increase cell tolerance to high concentrations of 3-HP, compared with the WT strain” (Line 627-628).

l. 563: isomerized relation???

We are sorry that we used inappropriate description. We have now revised it as “(a) Candidate enzymes that could potentially be able to degrade 3-HP.” (line647-648).

l. 567-568: I do not understand the title of this figure.

We are sorry for the confusion. The title of this figure has now been revised to “The final 3-HP production and its conversion to acrylic acid.” (Line 652).

Reviewer #3 (Remarks to the Author):

This paper describes the high production of 3-hydroxypropionic acid (3-HP), the critical precursor for acrylic acid. The authors introduced and deleted various proteins, including modifying several intracellular pathways, and higher CO₂ fixation, leading to a significant increase in the 3-HP.

This study is highly appreciated since it has succeeded in producing a quantity of 3HP that far surpasses known reports through careful planning and extensive experimentation. However, the following points should be revised or addressed.

Major comments.

1. Some proteins should be characterized as those proteins function as the authors proposed. It is unclear whether SUL1 imports bicarbonate or production increase results from the side effect of SUL1 protein, like sulfate import. Efflux activity of 3-HP by ESBP6 was not also demonstrated. Growth enhancement might be observed by its other effect. Using NaH¹³CO₃ and 3-HP, cellular transport would be possible to characterize those transporters. Also, for UGA1, its biochemical characterization or analysis of metabolic profile alteration by deleting uga1, which might show the increase or decrease of the proposed intermediates or other metabolites, would be desirable. Those analyses would strengthen this study as the comprehensive modifications, including substrate import, product efflux, and inhibition of intermediate degradation, benefit the high production of valuable metabolites.

Response: We thank the reviewer for these suggestions. These suggestions helped us to improve the manuscript. Sul1 and Sul2 both belong to the sulfate transport protein family. Yet, from our results overexpression of Sul2 did not improve 3-HP production as Sul1 did. We thus used AlphaFold to simulate possible structures of yeast Sul1 and Sul2, which share a close phylogenetic relationship with human SLC26A11 (Fig. S2), a Cl⁻/HCO₃⁻ exchanger in the kidney. While both Sul1 and Sul2 structures exhibited significant similarity to SLC26A11, Sul1 displayed a higher number of equivalent positions, suggesting a higher possibility of having a bicarbonate transport function compared with Sul2. We have now also toned down the descriptions of Sul1 as a potential bicarbonate importer.

Figure S2. Superimposed structures of SLC26A11 with Sul1 (a) and Sul2 (b). Protein structures for human SLC26A11 (Uniprot ID Q86WA9), yeast Sul1 (Uniprot ID P38359), and yeast Sul2 (Uniprot ID Q12325) were obtained from AlphaFoldDB (<https://alphafold.ebi.ac.uk/>), and structure similarity was assessed using FATCAT (<https://fatcat.godziklab.org>).

Meanwhile, to prove the efflux function of *Esbp6*, we have now also tested the intracellular 3-HP concentrations in strains with or without *ESBP6* overexpression (Fig. S6). The results showed that the strain with the overexpression of *ESBP6* have a lower intracellular 3-HP concentration compared with that of the WT strain. This result together with the cell growth recover in the high 3-HP concentration medium indicated that *Esbp6* could potentially help exporting 3-HP out of the cell. We have now also toned down the descriptions of Sul1 as a potential bicarbonate importer.

Figure S6. Effects of *ESBP6* gene on intracellular 3-HP concentration. The intracellular 3-HP content was measured after the treatment of strains with 50 g/L 3-HP for 24 hours to assess the impact of the *ESBP6* gene. All data points were represented as the mean \pm standard deviation (SD) of biological triplicates.

For Uga1, we have now checked the metabolic profile of strains with or without *UGA1* deletion, and demonstrated that the deletion of *UGA1* almost abolished the generation of the proposed intermediate β -alanine from MSA degradation (Fig. S10). This result further demonstrated the potential function of Uga1 in MSA degradation.

Figure S10. Effects of *UGA1* gene on β -alanine concentration in QLW69 and QLW71.

2. As for Nile red staining (Fig. 2g), quantitative data, such as the diameter and the number of oil bodies, should be shown.

Response: We totally agree with the reviewer and thank for the suggestion. We have now added the Nile red quantification data in Fig. 2h. The new results showed that the relative fluorescence unit of Nile red (stain of lipid droplet) in strain QLW36 with up-regulated β -oxidation pathway and downregulated fatty acid biosynthesis pathway was significantly decreased compared with that of the control strain QLW26.

Fig. 2h. The fluorescence intensity of neutral lipids stained with Nile red in QLW26 and QLW36 was quantified in Relative Fluorescence Units (RFU).

Minor comments.

1. L. 136. Reference 32 mainly describes *SHST1* from *Stylosanthes hamata*, less statement of *sul1/2*. For *sul1/2*, other papers would be appropriate.

Response: We are sorry for the typo on this. The right reference of *Sul1/2* has now been added (Line 141).

2. L. 149-151. The data that acetate accumulation should be shown as a supplement.

Response: We agree with the reviewer, and we have now added the data of acetate accumulation in Fig. S5 (Line 160).

Figure S5. Acetate accumulation was observed after deleting the *CIT2* on QLW10. The control strain was QLW8 and QLW10, and the *CIT2* deleted strain was QLW11.

3. L. 158. Please add the reference for *Yhm2*

Response: Thanks for the suggestion. The reference for *Yhm2* has now been added (Line 168).

4. L. 184-188. Please add the references for *Ora1* and *ydfG*.

Response: Thanks for the suggestion. The references for *Ora1* and *YdfG* have now been added (Line 195-196).

5. L. 200-202. Please add the reference for *POX1* and *POX2* and explain those functions. Adding peroxisome in Fig. 2e might be desirable if these oxidations occur in the peroxisome. The slight effect of *POX1* and *POX2* might be due to the bottleneck of peroxisomal transport of metabolites.

Response: We thank the reviewer for the suggestion. We have now added the reference for *Pox1* and *Pox2* and explained those functions as: "The resulting

the QLW33 strain overexpressing *POX1* that encodes fatty-acyl coenzyme A oxidase improved 3-HP production to 5.8 g/L. Similarly, the QLW34 strain overexpressing *POX2* that encodes 3-hydroxyacyl-CoA dehydrogenase and enoyl-CoA hydratase, improved 3-HP production to 5.92 g/L (Fig. 2f).” (Line 213-216). Moreover, we have now added the peroxisome in Fig. 2e.

Fig. 2 (e) Carbon flux rewiring from fatty acids to 3-HP. **(f)** Downregulation of *FAS1* combined with upregulation of *POX2* and *POX1* improved the production of 3-HP.

6. L. 236-239. Please add the reference of *SUR1*.

Response: Thanks for the suggestion. The reference of *SUR1* was added (Line 251).

7. L. 243. “and” might be “at”.

Response: Fixed.

8. L. 442-444. Please re-check the mobile phase; the authors did not use organic solvent for HPLC?

Response: Thanks for the suggestion, we have re-checked the mobile phase, we use the Aminex HPX-87H column, this column did not use organic solvent for HPLC.

9. Please re-check that the description of Fig. 2c is correct. *MCR-N*, malonate semialdehyde reductase; *MCR-C*, Malonyl-CoA reductase

Response: We are so sorry for the typos and we have now corrected it in Fig. 2c.

10. Fig. 2g legend. Please state which strain was used as the control.

Response: We are sorry that we did not present more clearly in the original version of the manuscript and thanks for the suggestion. We have now added the strain numbers on Fig. 2g and other places when necessary.

11. *Fig. S1. Please re-check the description in the figure. ARTp-STB5 and TEF1p-STB5 would be correct. In the legend and figure, an inverse description is shown.*

Response: We are so sorry for this careless mistake, and thanks for the suggestion. We have now revised it in Fig. S1 and the legend.

Reviewers' Comments:

Reviewer #1:

Remarks to the Author:

The authors well addressed the reviewers' comments and improved the manuscript.

Reviewer #2:

Remarks to the Author:

My comments and concerns have been adequately addressed in the revision.

Reviewer #3:

Remarks to the Author:

The authors did an excellent job with the revised version and clarified this reviewer's concerns. This manuscript looks clear, and the data are robust.